# EMBEDDING LEARNING FOR APPROXIMATING PERSON-SPECIFIC COGNITIVE SIMILARITY : FOCUSING ON MEDICAL IMAGES

## ABSTRACT

Metric learning is often applied in scenarios where labels are well-defined or where there is a ground truth for semantic similarity between data points. However, in expert domains such as medical data, where experts perceive features and similarities differently on an individual basis, modeling psychological embeddings at the individual level can be beneficial. Such embeddings can predict factors that influence behavior, such as individual uncertainty, and support personalized learning strategies. Despite this potential, the amount of person-specific behavioral data that can be collected through similarity behavior sampling is insufficient in most scenarios, making modeling individual cognitive embeddings challenging and underexplored. In this study, we proposed integrating supervised learning on small-scale similarity sampling data with unsupervised autoencoder-based manifold learning to approximate person-specific psychological embeddings with significantly improved similarity inference performance. We conducted a large-scale experiment with 121 clinical physicians, measured their cognitive similarities using medical image data, and implemented person-specific models. Our results demonstrate that even in complex expert domains, such as medical imaging, where cognitive similarity varies between individuals, person-specific psychological embeddings can be effectively approximated using limited behavioral data.

## 1 INTRODUCTION

Deep machine learning (ML) models can provide abstract-level information regarding the similarity between data through embedding (Liu et al., 2020b; Bengio et al., 2013; Mikolov et al., 2013). For instance, samples positioned close together in the embedding space can be interpreted as semantically similar, whereas those that are far apart are different. However, the actual similarity between the samples may not always be reflected in the learned embedding of the model. Metric learning is a method in which a model learns a function of the actual similarities and differences between samples and fits this function to low-dimensional embedding. In this context, the 'actual similarity' refers to a conventional metric that is defined externally. For example, in labeled datasets, a clear metric states that data points with the same label should be closer to each other than those with different labels. Even in the absence of labels, many datasets have commonly accepted metrics, such as the general perception that dogs are more similar to cats than snakes. Most metric-learning approaches operate under such conditions; thus, sufficient training data are available.

By contrast, we introduce a specialized metric learning problem that approximates individual-level psychological (cognitive) embeddings in scenarios where there are significant differences in similarity metrics depending on the individual (Schroff et al., 2015; Liu et al., 2017; Hosseini et al., 2018; Luo et al., 2003). This issue is particularly relevant in expert-driven fields such as medical data, where interpretations often differ. For example, while chest X-ray (CXR) images are structurally simple, their interpretation is highly complex (Delrue et al., 2011; Pham et al., 2021). Even among experienced physicians, the cognitive similarities or diagnoses of CXR images can vary widely (see Section 3.3) (Krupinski, 2010). Moreover, because labels are defined using partial data features, they may not align with the similarities assessed from a holistic perspective. For example, in CXR data, it may be necessary to differentiate between 'Male' and 'Female,' while also distinguishing be-

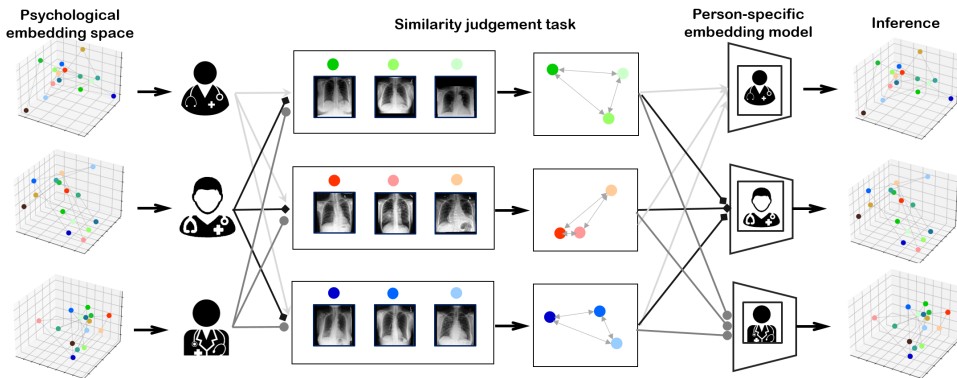

Figure 1: Illustration of the person-specific cognitive embedding modeling framework. Individual similarity measurement experiments arrange three images into a scalene triangle based on their pairwise similarities. This similarity data is then converted into triplets to train a person-specific embedding model. The resulting model approximates the person's psychological embeddings.

tween 'Normal' and 'Abnormal'. Therefore, in highly complex datasets, an individual's perception of similarity can remain independent of external metrics (labels).

This scenario is important because approximating individual-level cognitive metrics with psychological embeddings enables the inference of features considered more important by each person. Personalized learning strategies can be developed by identifying areas with high uncertainty because similar data points often exhibit similar uncertainties (Liu et al., 2020a; Mukhoti et al., 2021; Sanchez et al., 2022). This approach can also improve expert AI by transferring the psychological representations of superior experts to models.

Therefore, it is important to attempt to approximate cognitive embedding at the individual level; however, such attempts are rare. The biggest challenge lies in the inherent noise and difficulty of obtaining sufficient behavioral sampling for individual-level psychological embedding modeling (Molenaar & Campbell, 2009). In practical scenarios, modeling individual-level psychological embeddings requires overcoming the issue of insufficient behavioral sampling. As an alternative, we demonstrated that by integrating a loss function that learns from behavioral sampling data collected from individuals into an autoencoder-based framework, it is possible to synergistically achieve person-specific psychological embedding modeling that represents cognitive similarity, even with a limited amount of training data.

Our objective is not to improve metric learning algorithms or optimize models but to propose and experimentally validate a practical approach for applying metric learning to individual embedding learning. Specifically, we conducted a first-ever behavioral sampling experiment to measure the cognitive similarity of actual CXR images with 121 clinical physicians, focusing on realistic scenarios. After confirming significant variations in the cognitive similarity patterns across individuals, we implemented autoencoder-based models to represent each person's similarity metric at the embedding level (Fig. 1). The performance of the model was evaluated using individual behavioral data, and the robustness of our hypothesis was validated through ablation studies and simulations.

The key contributions of our study are as follows:

1) This is the first expert-based experimental study to model individual-level psychological embeddings, demonstrating the applicability of our approach to a realistic scenario using actual clinical physicians and medical data. 2) We showed that autoencoders can synergistically complement the limitations of cognitive similarity sampling in individual-level cognitive similarity approximations. (Proposing a new application scenario for autoencoders) Additionally, we empirically demonstrated the utility of the variable triplet loss, which was proposed to learn the psychological embeddings in the bottleneck layer of the autoencoder. 3) The experiment, which involved 121 physicians and medical imaging data, required considerable time and effort. The raw experimental data will contribute to research in fields such as human–AI collaboration.

## 2 Related Work

### 2.1 Metric learning

This study is closely related to metric learning in ML, which involves training models to learn the similarity between samples (Weinberger & Saul, 2009; Xing et al., 2002; Weinberger et al., 2005). Early algorithms focused on discriminating similar and dissimilar samples or with pre-defined metrics (Aherne et al., 1998; Elgammal et al., 2003). Recent approaches aims to learn the distance function in the embedding space of the model, representing similar and dissimilar samples as close and distant, respectively. For instance, selecting a reference image (anchor) among three data points helps determine which one is closer or farther from the reference, referred to as triplets (Hoffer & Ailon, 2015; Wang et al., 2017; Le-Khac et al., 2020; Ge, 2018; Hoffer & Ailon, 2015; Kim et al., 2020). Consequently, the triplet loss that reflects the conditions of these triplets in embedding space can be defined. It increases the distance between negative pairs more than the positive pairs. From the perspective of metric learning, the goal of our work is not to improve existing algorithms but rather to apply metric learning to model person-specific psychological metrics, which are not defined externally. This distinguishes our work from other studies.

### 2.2 Person-specific cognitive similarity modeling

Human inference operates through cognitive mechanisms that can be conceptualized as a hypothetical representational space, analogous to embeddings in machine learning models. In cognitive science, this representational framework is referred to as a psychological embedding. According to the theory of similarity-representation duality, embedding models trained on an individual's cognitive similarity metrics can concurrently encapsulate their cognitive features (Roads & Love, 2024). This framework suggests that similarity metrics specific to an individual can uncover the relative weighting of features involved in their perceptual processing of objects. By harnessing these insights, personalized embedding models have the potential to identify optimal learning domains and highlight knowledge deficits in experts (Cha & Lee, 2021).

Previous research in cognitive science has investigated tasks pertinent to the development of individualized embeddings. These efforts include systematic evaluations of human perceptual characteristics (Zhang et al., 2018), fine-tuning neural networks to enhance predictions of human similarity judgments (Tarigopula et al., 2023), and integrating neurological signals with cognitive embeddings (Palazzo et al., 2020). Although substantial progress has been achieved in collecting behavioral data for psychological embeddings across diverse stimuli (Hebart et al., 2020; Nosofsky et al., 2018; Wilber et al., 2014), relatively few studies have addressed the technical challenges associated with insufficient sampling for individual-level modeling. To address this gap, our study investigates the potential of leveraging unsupervised learning techniques, specifically autoencoders, to amplify limited person-specific similarity information.

In contrast to conventional metric learning frameworks that rely on predefined, consensus-based similarity metrics, person-specific metric learning becomes particularly valuable in contexts such as expert-driven domains, where cognitive similarity metrics exhibit significant inter-individual variation. Nevertheless, most prior study in this area has relied on benchmark datasets characterized by relatively minor differences in individual similarity perceptions (Peterson et al., 2018). Our work presents a novel contribution by validating the feasibility of individualized embedding modeling in real-world, professional datasets such as medical imaging, thereby bridging theoretical cognitive science and practical expert applications.

### 2.3 Autoencoder and manifold learning

Autoencoders learn the latent representations of data points in the bottleneck layer between the encoder and decoder by minimizing the reconstruction loss of the decoder for the input data (Berahmand et al., 2024; Tschannen et al., 2018; Wang et al., 2014). The objective is to learn latent representations and thereby discover hidden structures in the data. From the perspective of the manifold theory, training an autoencoder is equivalent to determining the parameters of a data manifold (Lempitsky, 2019; Lu et al., 2019). However, manifold structures are not always singular and it is common for data to belong to multiple manifolds (Hettiarachchi & Peters, 2015). Although the robustness of autoencoders has been demonstrated in numerous studies, explaining the local structure

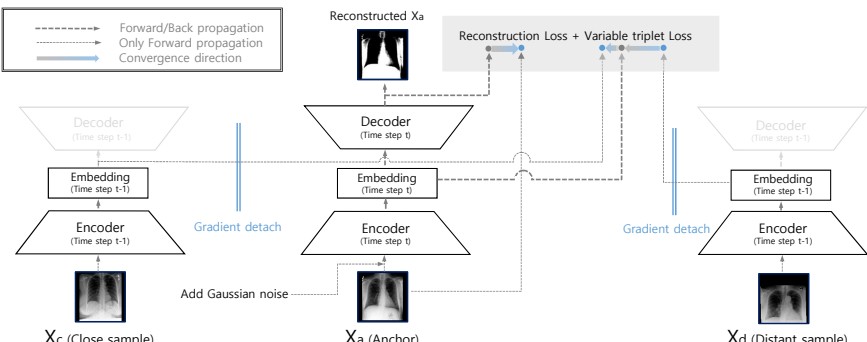

Figure 2: Illustration of the loss structure and its optimization process on a synthetic example.

of the manifolds represented in the embedding space learned by unconstrained autoencoders remains challenging (Tschannen et al., 2018). Instead, autoencoders can potentially learn various manifold structures depending on the initial weights of the model or certain constraints. If we assume that each individual's psychological embedding space represents a manifold, autoencoders can offer a useful guide for modeling person-specific psychological embeddings. However, this approach has not been explored previously.

## 3 APPROACH

### 3.1 COLLECTING SIMILARITY JUDGEMENTS

Sampling behavioral data for person-specific modeling may entail several limitations. First, behavioral data from a subject can typically only be used to train an individual model for that subject. Considering the laborious nature of behavioral sampling, the collection of triplets can incur high costs. Consequently, the amount of behavioral data that can be gathered from a single individual may be limited. Second, although the dimensionality of the embedding space is lower compared to the stimulus (i.e., raw images) space, it remains high-dimensional, posing challenges in case of intuitive handling (Roads & Love, 2024). This dimensionality can introduce significant noise when measuring similarity between data points. We implemented a triangular measurement framework treating each data point composing a triplet as an independent anchor to collect efficient behavioral data with minimal sampling and low uncertainty. Considering three data points, the selection of one of them as the anchor forms a unmeasured triplet $\mathcal{T} = (\mathbf{x}_a, \mathbf{x}_1, \mathbf{x}_2)$, where $\mathbf{x}_a$ denotes the anchor point, and $\mathbf{x}_1$ and $\mathbf{x}_2$ denote the other points, respectively. When the subject $\mathbf{s}$ selects a data point that is either more similar or dissimilar to $\mathbf{x}_a$ from $\mathbf{x}_1$ and $\mathbf{x}_2$, the $\mathcal{T}$ is expressed as the 'measured triplet' $\mathcal{T}^{\mathbf{s}} = (\mathbf{x}_a, \mathbf{x}_c, \mathbf{x}_d)$, where $\mathbf{x}_c$ and $\mathbf{x}_d$ denote the close and distant points, respectively, based on the behavioral measurements of the subject $\mathbf{s}$. Therefore, each measured triplet has two possible labels. Excluding the anchor, the remaining two images in a triplet are labeled as one close ("close" sample) and one relatively distant ("distant" sample). In our measurement procedure, three images were presented to the participant without specifying any anchors. Subsequently, the subject arranged the positions of these images in a scalene triangle shape, reflecting the degree of closeness or distance between each pair of images. Upon fixing the anchor image, a triplet was automatically determined with the remaining two images. Therefore, with three images presented in one instance, three triplets were collected. Thus, considering three data points sampled from the entire dataset without duplication $m$ times, $m$ measurement experiment instances can be conducted, thereby yielding $3m$ triplets. Although the triplets obtained from a single instance may be interdependent, this approach could regularize subject response by reducing the degree of freedom in representing cognitive distances between data points. Moreover it facilitates the collection of three sets of behavioral data from a single experiment, thereby enhancing sampling efficiency.

### 3.2 CONVOLUTIONAL AUTOENCODER WITH VARIABLE TRIPLET LOSS

Cognitive representation may rely on biological parameters such as neurological architecture (Kriegeskorte, 2015; Kubilius et al., 2018), suggesting the importance of considering suitable ML

model architectures and loss functions. However, to the best of our knowledge, a generally applicable ML architecture that can emulating psychological embeddings has not been proposed. Therefore, we empirically designed the embedding model architecture based on convolutional autoencoder (Chen et al., 2017) considering the context of medical image modeling without labels. Its architecture offers several advantages for our objectives. First, convolutional neural networks (CNN) are considered viable for modeling visual perception to date (Zeiler & Fergus, 2014). Second, the collection of sufficient data for training deep CNNs with numerous parameters through individual behavioral sampling can pose challenges. Autoencoders (unsupervised embedding learning methods) can partially address the insufficient behavioral data issue by directly extracting a meaningful feature independent of classes (Yang et al., 2022; Psenka et al., 2024; Bank et al., 2023).

The proposed autoencoder model comprises an encoder $\mathbb{E}(\cdot)_t$ and decoder $\mathbb{D}(\cdot)_t$, each comprising convolution and deconvolution layers, respectively, where $t$ denotes training iteration step (i.e., epoch). The fully connected bottleneck layer, serving as the output of the encoder and input of the decoder, was designed to approximate the region of interest, psychological embeddings. Consider images $\{\mathbf{x}, \cdots\}$ and the corresponding embeddings output of the encoder $\{\mathbb{E}(\mathbf{x})_t, \cdots\}$, where $\mathbb{E}(\mathbf{x}) \in \mathbb{R}^D$ denotes the $D$-dimensional vector. At each training iteration, we randomly sampled a batch of measured triplets among the training set (Refer to Sec. 3.4). Training aims to prompt an increase in the Euclidean distance between $\mathbb{E}(\mathbf{x}_d)_t$ and $\mathbb{E}(\mathbf{x}_a)_t$ compared to the distance between $\mathbb{E}(\mathbf{x}_c)_t$ and $\mathbb{E}(\mathbf{x}_a)_t$, while the autoencoder reconstructs the input $\mathbf{x}_a$. Thus, we aimed to minimize the following loss function for each subject $\mathbf{s}$'s model at iteration $t$.

$$\mathcal{L}_{tri}(\mathcal{T}^{\mathbf{s}})_t = \underbrace{\|\mathbb{D}(\mathbb{E}(\mathbf{x}_a + \epsilon)_t)_t - \mathbf{x}_a\|_2}_{\text{recontruction loss}} + \underbrace{\alpha\|\mathbb{E}(\mathbf{x}_a)_t - \widehat{\mathbb{E}(\mathbf{x}_c)_{t-1}}\|_2 + \beta\|\mathbb{E}(\mathbf{x}_a)_t - \widehat{\mathbb{E}(\mathbf{x}_d)_{t-1}}\|_2}_{\text{variable triplet loss}},$$

(1)

where $\|\cdot\|_2$ denotes L2-norm, $\epsilon$ is Gaussian random noise, $\alpha$ and $\beta$ are hyperparameters that satisfy $\alpha > \beta$, and $\widehat{\cdot}$ denotes a constant tensor with no gradient flow. Further, $\mathbb{E}(\mathbf{x}_c)_{t-1}$ and $\mathbb{E}(\mathbf{x}_d)_{t-1}$ function as candidate vectors for embedding the subject for $\mathbf{x}_c$ and $\mathbf{x}_d$, respectively, indicating the convergence target of $\mathbf{x}_a$ at iteration $t$. However, with training progression, the candidate vectors for embedding change, rendering the overall convergence target of the loss function variable. A weakening in the convergence of stochastic gradient descent is anticipated, but with properly selected hyperparameters like learning rate, batch size, $\alpha$, and $\beta$, optimization primarily depends on reconstruction loss, minimizing significant convergence issues. Although autoencoders can effectively learn compressed representations, they are prone to overfitting and encounter challenges when determining feature importance (Meng et al., 2017). The variable triplet loss can be interpreted as constraints guiding the training of autoencoders towards the identification of features that are more specific to the target individual among the candidate features they can explore (Fig. 2).

## 3.3 PERSON-SPECIFIC SIMILARITY PATTERN QUANTIFICATION

To quantitatively express and compare the cognitive similarities among subjects participating in experiments for the same dataset, we defined **similarity pattern vector (SPV)**. Assume that we collected behavioral datasets through multiple instances $m$ times from subject $\mathbf{s}$. This formed a set $\mathsf{T}_\Omega^{\mathbf{s}}$ comprising $3m$ triplets. Each element triplet $\mathcal{T}_i^{\mathbf{s}}$ ($\forall i \in \{1, \cdots, 3m\}$) composing $\mathsf{T}_\Omega^{\mathbf{s}}$ can be transformed into binary labeling. Therefore, assuming each triplet as an independent dimension determining the similarity pattern of the subject $\mathbf{s}$, the similarity pattern is defined as follows.

$$SPV(\mathsf{T}_\Omega^{\mathbf{s}}) = [\mathbb{O}(\mathcal{T}_1^{\mathbf{s}}), \mathbb{O}(\mathcal{T}_2^{\mathbf{s}}), \cdots, \mathbb{O}(\mathcal{T}_{3m}^{\mathbf{s}})],$$

(2)

where $\mathbb{O}(\mathcal{T}_{\mathbf{n}}^{\mathbf{s}})$ denotes one of the possible similarity relationships for the $\mathbf{n}$-th $\mathcal{T}^{\mathbf{s}}$, expressed as 1 for one relation and 0 for another. Thus, $SPV(\mathsf{T}_\Omega^{\mathbf{s}})$ is the $3m$-dimensional one-hot vector expression, which indicates the person-specific similarity pattern.

## 3.4 TRAINING AND EVALUATION OF THE MODEL

The triplet set $\mathsf{T}_\Omega^{\mathbf{s}}$ was randomly assigned to the training set $\mathsf{T}_{\mathbf{T}}^{\mathbf{s}}$, validation set $\mathsf{T}_{\mathbf{V}}^{\mathbf{s}}$, and evaluation set $\mathsf{T}_{\mathbf{E}}^{\mathbf{s}}$ according to a predetermined ratio. While optimizing Eq. 1 using $\mathsf{T}_{\mathbf{T}}^{\mathbf{s}}$, the optimal model was determined considering the highest inference accuracy achieved on $\mathsf{T}_{\mathbf{V}}^{\mathbf{s}}$. Further, to address the scenario wherein inference accuracy must be treated across various models, we defined a predictive

evaluation function $\mathcal{F}$ for single triplet $\mathcal{T}_i^s$ as follows.

$$\mathcal{F}\left(\mathbb{E}_j(\mathbf{x}), \mathcal{T}_i^s\right) = \begin{cases} 1, & \text{if } \|\mathbb{E}_j(\mathbf{x}_a) - \mathbb{E}_j(\mathbf{x}_d)\|_2 > \|\mathbb{E}_j(\mathbf{x}_a) - \mathbb{E}_j(\mathbf{x}_c)\|_2 \\ 0, & \text{otherwise} \end{cases}, \qquad (3)$$

where, $\mathbb{E}_j(\mathbf{x})$ is output over fed $\mathbf{x}$ of the encoder trained using $\mathsf{T}_{\mathbf{T}}^{\mathbf{j}}$ and $\mathsf{T}_{\mathbf{V}}^{\mathbf{j}}$, $(\mathbf{x}_a, \mathbf{x}_c, \mathbf{x}_d)$ is $\mathcal{T}_i^s$.

In case of considerable diversity in cognitive patterns among subjects, (Sec.4.3) a model trained on $\mathsf{T}_{\mathbf{T}}^{\mathbf{i}}$ is strongly expected to achieve higher performance on $\mathsf{T}_{\mathbf{E}}^{\mathbf{i}}$. Whereas, the performance on $\mathsf{T}_{\mathbf{E}}^{j}$, (for $j \neq \mathbf{i}$), should be lower compared to $\mathsf{T}_{\mathbf{E}}^{\mathbf{i}}$. We defined the performance measured on the evaluation set comprising $h$ triplets obtained from the target subject $\mathbf{j}$ of the model as **Specific Performance (SP)** : $(\Sigma_{i=1}^h \mathcal{F}(\mathbb{E}_j(\mathbf{x}), \mathcal{T}_i^s))/h \times 100$ $(i = 1, \cdots h)$. Further, the performance measured on the test set collected from all $g$ subjects measured with the $\mathsf{T}_{\mathbf{E}}$ except the target subject $\mathbf{j}$ of the model was defined as **Non-Specific Performance (NSP)** : $(\Sigma_{k=1}^g \Sigma_{i=1}^h \mathcal{F}(\mathbb{E}_k(\mathbf{x}), \mathcal{T}_i^s))/h(g-1) \times 100$ $(i = 1, \cdots h, k = 1, \cdots g, k \neq \mathbf{j})$. Note that the reliability of NSP improves with a larger experimental group size since NSP depends on the subject group.

### 3.5 Qualitative Analysis

The performance of the embedding model can be qualitatively evaluated by comparing the locations of the top $n$ pixels predicted by the model to influence similarity judgments with the $n$ pixels identified by experienced clinicians. The annotation process of the model begins by selecting a reference unit with the highest variance among the embedding outputs of a separate reference dataset. Once the reference unit is determined, uniform noise is iteratively added to each pixel of the test image, which is then input into the model to compute the variance in the reference unit's output. This process is conducted individually for each pixel (Sec. A.6). Subsequently, the top $n$ pixels that caused the greatest variance in the reference unit's output due to the added noise are identified and compared with the pixels annotated by expert clinicians.

## 4 Experiments

Herein, we describe the human behavioral experiment setup and dataset (Sec. 4.1.-4.2) and present evidence that the similarity perception patterns of physicians vary on an individual basis (Sec 4.3). Then we evaluate the predictive performance of embedding models trained on a person-specific basis (Sec 4.5). We provide evidence that the performance of embedding models can improve with the scale of behavioral sampling through human surrogate model simulations (Sec. 4.7). Additionally, we examine the significance of each term of loss function through ablation study (Sec. 4.8).

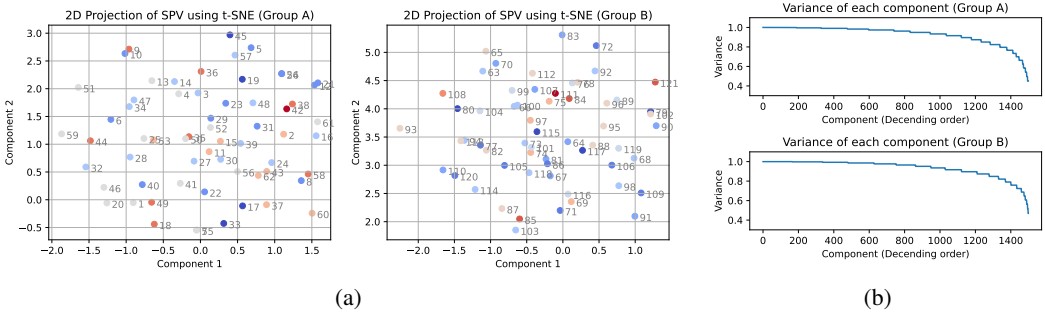

(a)                                                                                    (b)

Figure 3: Results of group-based similarity pattern analysis for all subjects. (a) 2D t-SNE visualization of similarity pattern vectors (SPVs). The colors represent the results of a separate image interpretation test conducted with the subjects, where red indicates relatively higher performance. (b) Variance of components in the SPVs (Decending order).

### 4.1 Datasets

CXR images serve as a crucial diagnostic modality in all clinical fields owing to their capability to contain wide clinical information. Moreover, interpreting CXR images can be challenging even

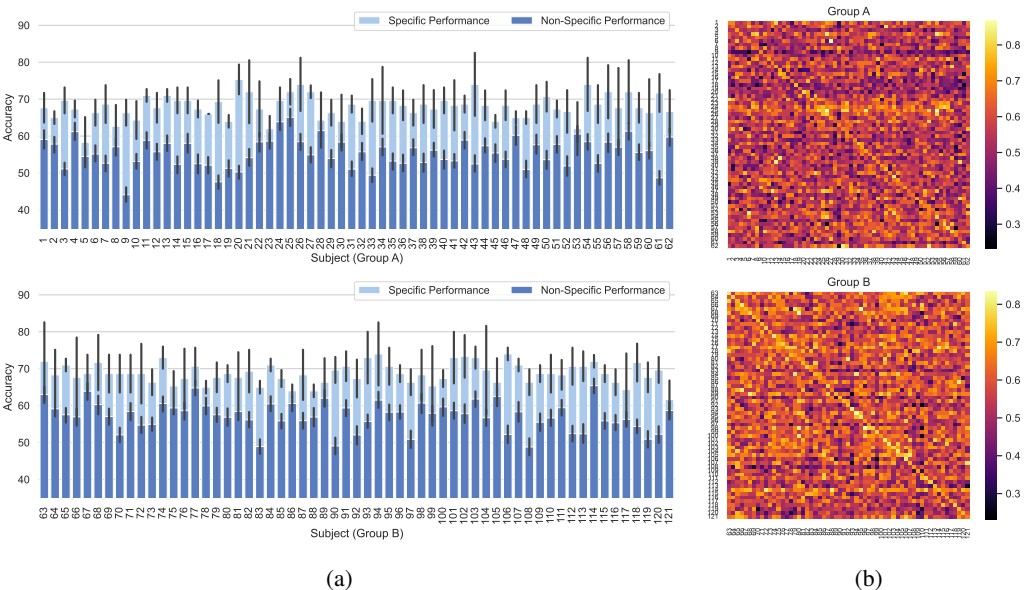

(a)                                                        (b)

Figure 4: Similarity inference performance analysis. (a) Model-specific results (Accuracy: %, bar: standard deviation). (b) Cross-performance heatmap (X-axis: subject triplets, Y-axis: models).

for experienced clinicians, and they may have diverse similarity perception patterns. This scenario aligns well with our task objectives. To compensate the limitations of single-domain experiments, we constructed two different experimental datasets. The CXR-A dataset was formed via random sampling of images labeled as 'Normal' or 'Abnormal' from the CheXpert 1.0 dataset (Irvin et al., 2019), whereas the CXR-B dataset comprised images labeled as 'Edema' or 'Pneumonia' from the same source. In our experiments, labels were unnecessary; however, we used this approach to sample the two sub-datasets from different distributions. Each dataset comprised 500 subsets, with each subset comprising three images (Therefore, resulting $\mathcal{T}^{\Omega}$ comprised 1500 $\mathcal{T}$ after the subject similarity measurements). CXR-A images might not clearly show lesions, leading physicians to focus on overall anatomical outlines, whereas CXR-B images, with more evident lesions, may lead physicians to focus on pathology (Behzadi-Khormouji et al., 2020; Homayounieh et al., 2021).

### 4.2 HUMAN EXPERIMENTS FOR SIMILARITY MEASUREMENT

We conducted experiments with 121 government-licensed clinical physicians, recruited from the official medical association. After randomly selecting 150 subjects, 121 were included, excluding those who withdrew. Subjects were assigned to Group A (1-62) and Group B (63-121), with each group using CXR-A and CXR-B, respectively. In the experiment, three unlabeled images (1 subset) were shown on the monitor, and subjects were asked to drag balls to indicate similarity, with closer balls representing more similarity. Despite a flexible scale between ball distances, the subjects were requested to maintain consistent criteria across all instances. The experiment had no time limit, and breaks were allowed. Group A took an average of 304 minutes to complete 500 experiments, while Group B took 245 minutes. Subsets were presented in a randomized order.

### 4.3 SIMILARITY PATTERN COMPARISON

Fig.3 (a) shows the t-SNE (Van der Maaten & Hinton, 2008) dimensionality reduction of the SPV (Eq.2) from the all subjects over belonging group. Each component of SPV was formed by all 1500 triplets collected from each subject. The similarity patterns among subjects from both groups were diverse, and did not form clusters. In the multivariate runs test conducted to assess randomness, SPV demonstrated randomness with p-values of 0.18 and 0.07 for the CXR-A and CXR-B groups, respectively. Thus, individuality embedding modeling for each person is necessary. Fig. 3 (b) illustrates the variance of each component of the the SPV across all subjects (descending order).

Despite the high variance in most components, the presence of components with relatively low variance suggested the potential of partial similarity patterns generally shared by subjects.

## 4.4 MODELING DETAIL

The best model architecture designed herein comprised an encoder with four convolutional layers, and a decoder with four deconvolutional layers (See A.2). All results, except for the comparative experiment (ablation study) presented in Sec. 4.8, are reported based on the best architecture. The embedding layer (region of interest), was a one-dimensional tensor with 64 units. Original gray images of size $1024 \times 1024$ presented to subjects were reduced to ($1 \times 128 \times 128$) and employed as model input data. We employed the PyTorch (Paszke et al.) for all experiments. We fixed the learning rate to $10^{-4}$ and used the Adam optimizer (Kingma & Ba, 2014) with 32 mini-batch size. The hyperparameters for training were consistent across all models. $\alpha$ and $\beta$ were set to 1.2 and 1, respectively; however, they were fine-tuned by the algorithm to reflect the distance scale reported by the subjects during the experiment.

## 4.5 PERFORMANCE OF PERSON-SPECIFIC SIMILARITY INFERENCE (MAIN STUDY)

Fig. 4 (a) shows the SP and NSP of all individual models. The number of triplets for training, validation, and evaluation was randomly set to 1410, 60, and 30, respectively. The final performance was reported using 3-fold cross-validation with random selection. Our models achieved significantly higher SP (Group A average 68%, Group B average 68.7%) compared to the chance level, thereby validating the efficacy of our approach. Considering the diverse similarity patterns among subjects, models may exhibit lower predictive performance on evaluation triplets of different subjects.(Models should achieve specific strong performance for test triplets for corresponding subject) Notably, across all models, a consistent trend of the NSP being lower than SP as observed. Fig. 4 (b) presents the cross-predictive performance on a heat-map for person-specific models on the evaluation triplets for all subjects. While displaying distinct predictive tendencies specific to the target subject's evaluation data, the models demonstrated random patterns for the data of the non-corresponding subjects, thereby aligning with the similarity patterns in the group (Sec. A.5).

## 4.6 RESULTS OF QUALITATIVE ANALYSIS (SUB-STUDY)

Figure 5 illustrates examples of the top 10 image regions where changes in pixel values influenced similarity, as determined by the procedure described in Section 3.5. The annotations shown on the left image were performed by experienced clinicians. The regions affecting experts' similarity judgments differed between CXR-A and CXR-B. In CXR-A, similarity judgments were primarily influenced by anatomical structures such as bones and the thorax, whereas in CXR-B, regions around lung lesions played a key role. These tendencies were also reflected in the annotations generated by the proposed model. Table 1 summarizes the proportion of pixels, averaged across all subjects, where the top 10 similarity-determining pixels predicted by the model were within a 2-inch distance of the 10 pixels identified by experienced clinicians. The proposed model demonstrated patterns closely aligned with those of the experts, while the comparative model (a conventional autoencoder described in Section 4.8) exhibited significantly weaker predictive performance.

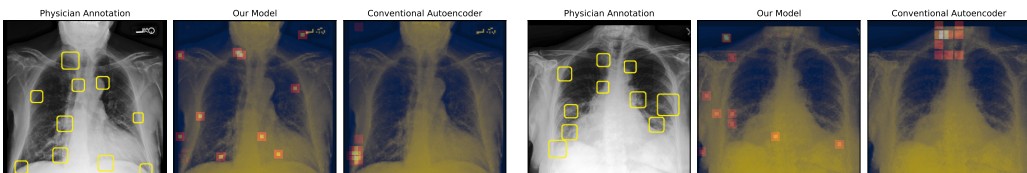

    (a) Example of CXR-A test image 1 (Subject 2)    (b) Example of CXR-B test image 1 (Subject 68)

Figure 5: Annotation of image regions where changes in pixel values impact embedding similarity.

Table 1: Proportion of top 10 similarity-determining pixels within 2 inches of expert annotations (CA : Conventional autoencoder).

| CXR-A Test Image 1 (n=62) | | CXR-A Test Image 2 (n=62) | | CXR-B Test Image 1 (n=59) | | CXR-B Test Image 2 (n=59) | |
|---|---|---|---|---|---|---|---|
| Our Model (%) | CA (%) | Our Model (%) | CA (%) | Our Model (%) | CA (%) | Our Model (%) | CA (%) |
| 49.3±17.1 | 13.8 ±12.7 | 50.8±10.9 | 14.8±9.7 | 51.0±17.2 | 22.8 ±12.3 | 45.4±14.5 | 18.9±11.8 |

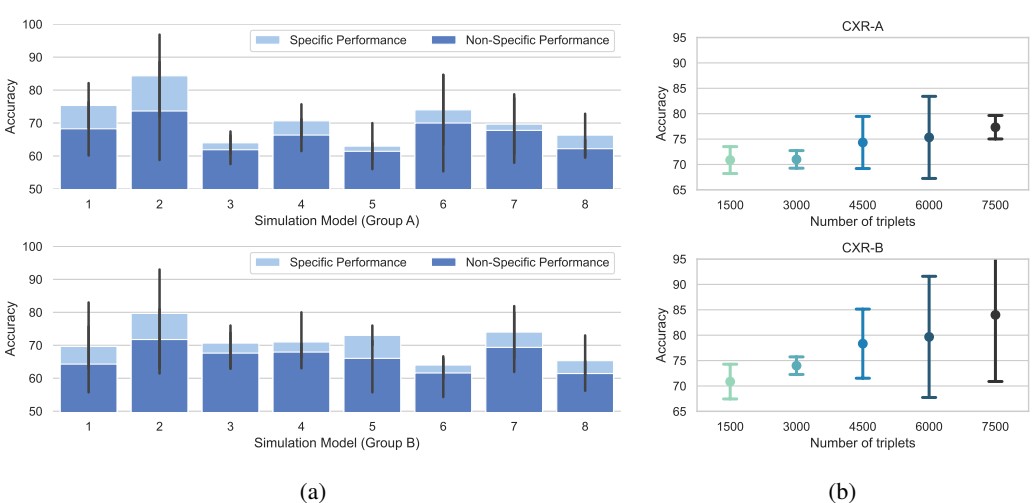

(a)                                             (b)

Figure 6: Inference performance of simulation experiments. (a) Model-specific results (Accuracy: %, bar: Standard deviation). (b) Effect of amount of triplets in simulation (bar: standard deviation).

### 4.7 SIMULATION EXPERIMENTS

We showed that the embeddings of the original model could be reconstructed in preserved similarity even in simulations wherein the subjects were replaced with ML models. We sampled 100 images each from CheXpert, which were not included in CXR-A or CXR-B. Thereafter, 16 different binary classification CNN models were trained for each set. We measured the Euclidean distance of output vectors inner layer immediately before the final layer for each model for the same triplets as in the human behavioral experiments to collect the similarity relationship. Subsequently, the secondary embedding model was trained in the same manner of the modeling of human embedding (Fig. 6 (a)). The performance was slightly higher than that in human behavioral experiments, indicating that noises may occur in the similarity measurement process for human subjects. Moreover, in the experiments wherein the number of subsets for CXR-A and CXR-B was increased using additional data, the performance of the embedding model for simulation exhibited an improvement according to the number of triplet samples for model training (Fig. 6 (b)). Thus, this implies increasing human behavioral experiment sample sizes could enhance embedding model performance in future.

### 4.8 ABLATION AND COMPARISON STUDY

This section summarizes the study designed to demonstrate the effects of the proposed modeling approach components on the loss function. Table 2 shows the inference performance of models trained by excluding each component (reconstruction and variable triplet losses) while maintaining the same training settings as in our proposed method. In both the ablated settings, the performance exceeded the chance level; however, it was lower than that of our proposed setting. An important finding is that the setup excluding variable triplet loss (i.e., conventional autoencoder) surpasses the opposite setting. It shows prioritizing reconstruction loss optimization offers advantages for embedding learning over exclusively optimizing triplet loss. This implied the presence of common latent similarity patterns among subjects that can be learned solely by autoencoders and indicated that the features that cannot be trained through triplet loss alone owing to sampling limitations were trained via the optimization of the reconstruction loss of the autoencoder. Moreover, in the ablation set-

Table 2: Results of ablation study (error: standard deviation).

| Methods | Group A (n=62) | | Group B (n=59) | |
|---|---|---|---|---|
| | SP (%) | NSP (%) | SP (%) | NSP (%) |
| Autoencoder with variable triplet loss (Our methods) | 68.0±3.2 | 55.5±3.9 | 68.8±2.7 | 57.2±3.8 |
| Variable triplet loss only (Excluding reconstruction loss) | 53.9±8.0 | 53.8±3.8 | 55.7±5.0 | 55.0±4.8 |
| Autoencoder only (Excluding variable triplet loss) | 57.9±6.0 | 56.8±3.8 | 62.3±4.8 | 59.7±3.1 |
| Encoder with triplet loss (Excluding decoder) | 61.8±3.1 | 56.3±2.8 | 60.7±4.5 | 57.0±4.1 |
| Encoder only (Excluding decoder and variable triplet loss) | 62.3±3.4 | 55.3±3.3 | 60.9±4.0 | 56.0±3.5 |

ting, SP was marginally higher than NSP; however, no significant superiority was observed, thereby suggesting an incapability to extract person-specific features. Meanwhile, an ablation experiment was conducted by adding a classifier to the encoder of the autoencoder, utilizing cross-entropy loss alongside variable triplet loss. In this ablation experiment aimed at validating the utility of the decoder, the performance was higher than that of the variable triplet loss-only setting; however, it did not surpass the performance of our proposed method. In summary, these evidences support our claim that variable triplet loss guides person-specific feature learning in autoencoders.

## 5 LIMITATION

While pioneering person-specific similarity-based cognitive embedding, this study faces limitations inherent to human behavioral experiments, such as uncertainty in similarity judgments and interdependent measurements. Future studies should adopt systematic experimental designs to address this noise. Additionally, our experiments focused on a limited set of neural network architectures and did not explore optimal hyperparameter tuning or provide theoretical proof for the hypothesis that person-specific cognitive similarity can guide autoencoder manifold learning.

## 6 CONCLUSION

This study proposed an autoencoder-based person-specific embedding modeling framework that approximated cognitive similarities between subjects in CXR data and conducted a large-scale behavioral experiment with clinical physicians. To the best of our knowledge, this is the first such study attempt. Specifically, we demonstrated that our approach can be applied in domains where significant inter-observer variability in similarity perception exists, such as in the complex interpretation of CXR images. As our experimental design did not include any domain-specific constraints or assumptions unique to the medical field, we believe that our method can potentially be generalized to other domains.

We hypothesized that individual psychological embeddings reflect features learned independently of external metrics (such as labels). According to this hypothesis, information derived from high-dimensional data that is cognitively interpreted may lie on a lower-dimensional psychological manifold. Autoencoders probabilistically learn the data manifold independent of external metrics. Therefore, autoencoders may offer a useful framework for approximating human-metric-independent embeddings. We further suggest that triplet loss, which captures individual similarity, may have acted as a perturbation that guided the autoencoder toward learning a specific manifold. However, theoretical proof is beyond the scope of this study and should be explored in future studies. Additionally, through simulations and ablation studies using person surrogate models, we confirmed the robustness of the proposed method and demonstrated the potential for proportional improvements in the model inference performance as the scale of behavioral data sampling increases.

Our study, which uses a multidisciplinary approach that integrates cognitive science, machine learning, and expert knowledge applications, demonstrates the potential of aligning deep neural networks with human representational mechanisms as a tool for understanding human cognitive representations. This approach could also potentially contribute to the development of machine-learning algorithms that support personalized learning for experts. Future research will aim to provide theoretical proof for the hypothesis that variable triplet loss can guide manifold learning in autoencoders and enhance the scalability of the proposed method by applying various learning algorithms suggested in the field of metric learning.

## 7 ETHICS STATEMENT

We review several ethical issues that may arise in this study. This study was approved by the Institutional Review Board (IRB No. removed) to conduct experiments involving human subjects using medical data. Participants were compensated with an amount exceeding the legally mandated minimum wage. To address ethical concerns, particularly those arising from involving the general public in experiments using medical data, we intentionally restricted the participants to qualified professionals. All participants voluntarily provided informed consent. The detailed statistics of the participants are summarized in the Appendix. The images used in this study (CXR-A and CXR-B) were obtained from the publicly available CheXpert dataset and their use was reviewed in advance by the IRB. We carefully considered the potential ethical issues that could arise during the advancement of this study. Nonetheless, technologies predicting human cognitive characteristics may pose ethical challenges as they could expose the vulnerabilities of professionals, compromise their judgment through adversarial attacks, or lead to adverse selection by clients. Therefore, as this study advances, it may be necessary to address ethical considerations simultaneously.

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

# A APPENDIX

## A.1 SUBJECT CHARACTERISTICS

The subjects were randomly assigned to either Group A or Group B. The mean ages of subjects in Groups A and B were 31 years (SD = 5.9 years) and 36.6 years (SD = 6.4 years), respectively. The number of female participants in Groups A and B were 3 and 6, respectively. Table 3 provides information for each participant, however, to protect participant anonymity, gender and specific clinical backgrounds are not indicated.

* GP1: General physician without clinical training, GP2: General physician with internship traning, SP1: Specialist in internal medicine, SP2: Specialist in chest X-ray-related disciplines (anesthesiology, thoracic surgery, occupational medicine, family medicine, pediatrics, emergency medicine, radiology, radiation oncology, nuclear medicine), SP3 : Other specialists (otorhinolaryngology, rehabilitation medicine, ophthalmology, orthopedic surgery, obstetrics and gynecology, neurology, dermatology, plastic surgery, neurosurgery)

## A.2 MODEL ARCHTECTURE

We implemented our model using Python version 3.8.18 and the PyTorch(Paszke et al.) version 2.2.1 library on the Ubuntu 18.0 environment. Please refer to Fig. 7 for the logical structure of the model. The random seed used for the final model selection and the conda virtual environment configuration are provided in the attached files.

## A.3 COMPUTER RESOURCES

All models were trained using a single RTX 3090 GPU. We were able to train 10 models simultaneously on a single GPU. Although our individual models are relatively small in scale, they must be trained separately for each subject. We utilized 4 GPUs concurrently to perform parallel computations for multiple models. Training the same model 20 times with different random seeds allowed us to select the optimal model. The time required to train a single model ranged from approximately 3.5 to 5.5 hours.

Table 3: Key information of the subjects

| | Group A | | | Group B | |
| --- | --- | --- | --- | --- | --- |
| Subject ID | Age | Clinical field* | Subject ID | Age | Clinical field* |
| 1 | 45 | SP2 | 63 | 38 | SP4 |
| 2 | 39 | SP2 | 64 | 44 | GP2 |
| 3 | 33 | SP2 | 65 | 38 | SP2 |
| 4 | 34 | SP2 | 66 | 36 | SP2 |
| 5 | 32 | SP2 | 67 | 39 | SP3 |
| 6 | 34 | SP3 | 68 | 37 | SP1 |
| 7 | 33 | SP3 | 69 | 39 | SP3 |
| 8 | 32 | SP3 | 70 | 40 | SP3 |
| 9 | 33 | GP1 | 71 | 34 | SP3 |
| 10 | 32 | SP2 | 72 | 36 | SP3 |
| 11 | 30 | SP2 | 73 | 34 | SP1 |
| 12 | 31 | SP2 | 74 | 33 | SP3 |
| 13 | 29 | SP2 | 75 | 43 | SP2 |
| 14 | 30 | SP2 | 76 | 38 | SP2 |
| 15 | 32 | SP3 | 77 | 32 | SP1 |
| 16 | 29 | SP2 | 78 | 36 | SP3 |
| 17 | 33 | SP2 | 79 | 35 | SP1 |
| 18 | 32 | GP2 | 80 | 36 | SP2 |
| 19 | 29 | GP2 | 81 | 36 | SP3 |
| 20 | 29 | GP2 | 82 | 41 | SP2 |
| 21 | 32 | GP1 | 83 | 35 | SP1 |
| 22 | 30 | GP1 | 84 | 35 | SP1 |
| 23 | 28 | GP1 | 85 | 32 | SP2 |
| 24 | 26 | GP2 | 86 | 32 | GP1 |
| 25 | 31 | GP1 | 87 | 32 | SP3 |
| 26 | 27 | GP2 | 88 | 31 | SP3 |
| 27 | 29 | GP2 | 89 | 30 | GP1 |
| 28 | 27 | GP2 | 90 | 30 | SP1 |
| 29 | 29 | SP2 | 91 | 32 | SP2 |
| 30 | 28 | GP1 | 92 | 38 | GP2 |
| 31 | 28 | GP1 | 93 | 36 | SP2 |
| 32 | 28 | GP1 | 94 | 35 | GP2 |
| 33 | 27 | GP2 | 95 | 34 | SP1 |
| 34 | 29 | GP1 | 96 | 30 | GP1 |
| 35 | 27 | GP2 | 97 | 30 | GP2 |
| 36 | 31 | GP1 | 98 | 33 | GP1 |
| 37 | 26 | GP1 | 99 | 41 | SP2 |
| 38 | 28 | GP2 | 100 | 32 | GP1 |
| 39 | 27 | GP2 | 101 | 33 | GP2 |
| 40 | 30 | GP1 | 102 | 32 | GP1 |
| 41 | 28 | GP1 | 103 | 32 | GP1 |
| 42 | 28 | GP2 | 104 | 37 | GP2 |
| 43 | 30 | GP1 | 105 | 26 | GP2 |
| 44 | 30 | GP1 | 106 | 30 | SP2 |
| 45 | 27 | GP2 | 107 | 27 | GP2 |
| 46 | 30 | GP1 | 108 | 26 | GP1 |
| 47 | 26 | GP2 | 109 | 38 | GP2 |
| 48 | 28 | GP1 | 110 | 26 | GP1 |
| 49 | 26 | GP2 | 111 | 52 | SP2 |
| 50 | 26 | GP1 | 112 | 58 | SP2 |
| 51 | 27 | GP2 | 113 | 47 | SP2 |
| 52 | 25 | GP1 | 114 | 44 | SP2 |
| 53 | 27 | SP3 | 115 | 45 | SP2 |
| 54 | 27 | GP2 | 116 | 46 | SP1 |
| 55 | 33 | SP3 | 117 | 44 | SP1 |
| 56 | 26 | GP1 | 118 | 46 | SP1 |
| 57 | 27 | GP1 | 119 | 44 | SP1 |
| 58 | 55 | SP1 | 120 | 42 | SP3 |
| 59 | 48 | SP2 | 121 | 40 | SP2 |
| 60 | 46 | SP2 | | | |
| 61 | 44 | SP2 | | | |
| 62 | 42 | SP2 | | | |

## A.4 MAIN EXPERIMENTS (SIMILARITY JUDGEMENTS)

This section describes the main human behavior experiment, focusing on the procedures presented to the participants. Each instance in the main experiment consists of a triplet, composed of three images. At the start of each instance(Fig. 8), the three images forming the triplet are displayed on the left side of the screen (Fig. 9). Each image is matched with a drag ball of a different color. The order of tasks and the arrangement of images within each task are randomized for each subject. In the center of the screen, the matched drag balls are initially arranged in an equilateral triangle.

Participants are asked to drag the balls such that the matched images are positioned closer together as their perceived similarity increases (Fig. 10). If two images are perceived as identical at embedding level, participants should place the corresponding balls in the same position. After submitting

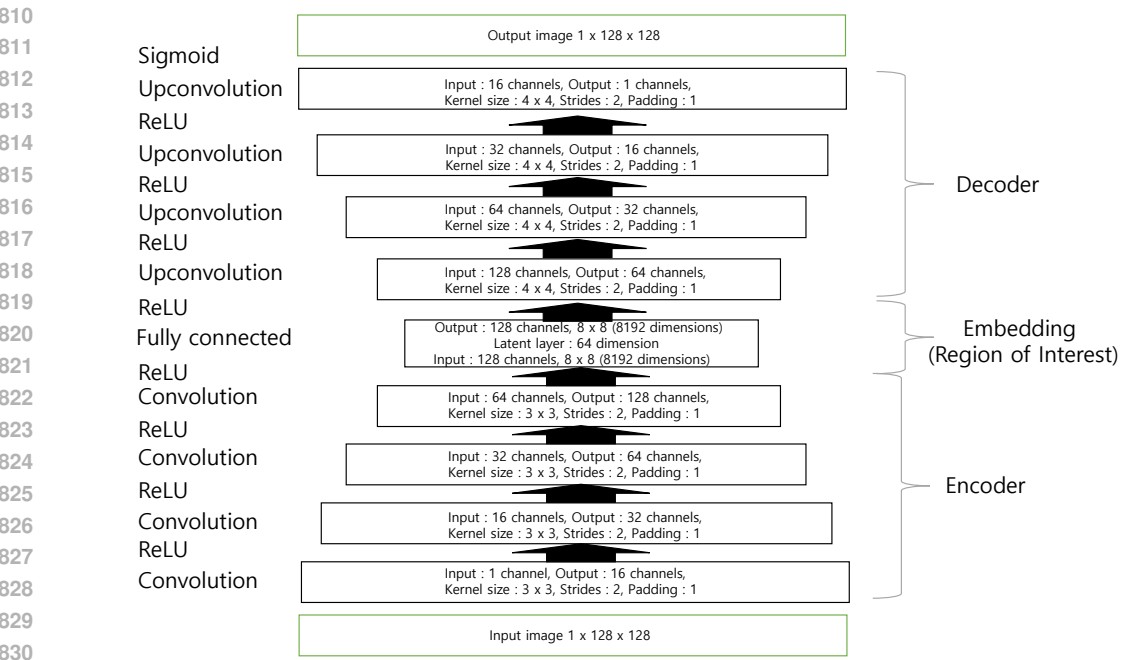

Figure 7: Architecture of our neural network model.

their similarity judgments, participants can input their confidence in the similarity between each pair of drag balls (Fig. 11). Clicking on the line connecting two drag balls changes its thickness, with thicker lines indicating higher confidence levels, measured on a 5-point scale. Although the confidence measurements are provided in the attachment files, they were not included in the analysis for this experiment.

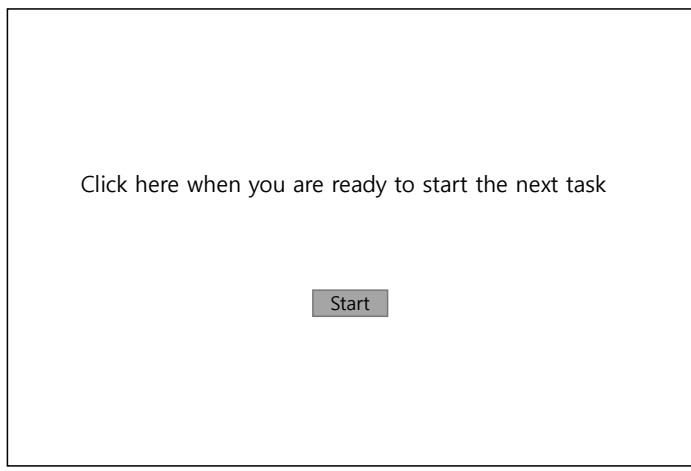

Figure 8: Screen image before start each task.

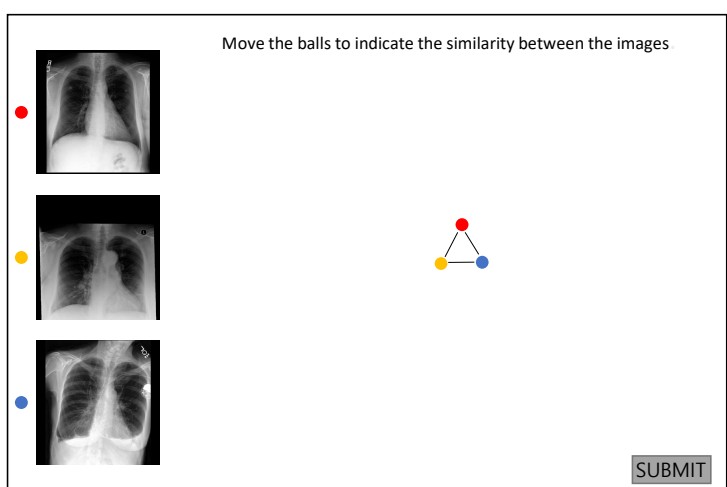

Figure 9: Initial task image.

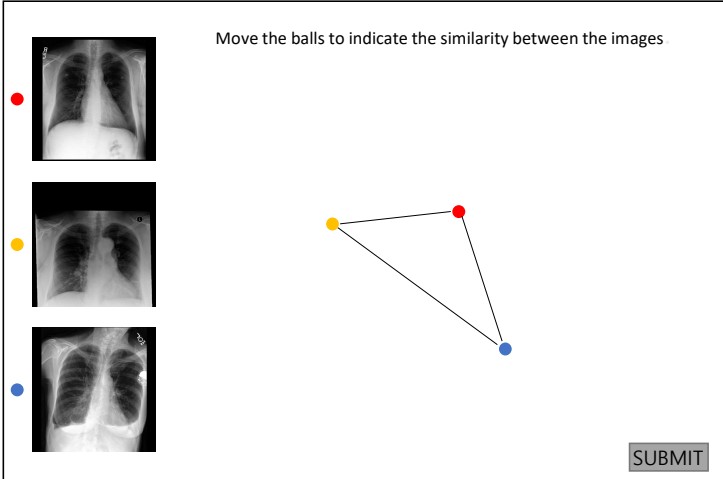

Figure 10: Subject's Ball Movement.

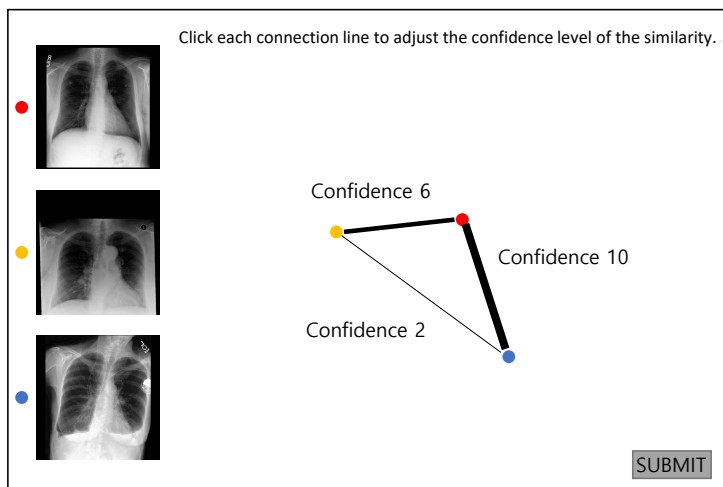

Figure 11: Input confidence level for each similarity judgements.

## A.5 Detail results of main study (similarity inference)

Tables 4 and 5 present the similarity inference performance of models generated in Group A and Group B, respectively, displaying Specific Performance (SP) and Non-Specific Performance (NSP) for each model. These tables include inference results for models trained using our proposed standard approach as well as two ablation settings. SP is reported as the mean and standard deviation of the performance across multiple models generated through cross-validation. However, it should be noted that the error presented for NSP is the standard error of the inference performance for all subjects except the targeted subject for the model. This distinction is made to clearly highlight the characteristics of the raw performance information used to calculate NSP across different subjects.

Table 4: Performance of each model for group A (Accuracy and Standard deviation: %)

| Subject ID | Our mothods | | Excluding reconstruction loss | | Excluding triplet loss | |
|---|---|---|---|---|---|---|
| | SP | NSP | SP | NSP | SP | NSP |
| 1 | 67.7±3.3 | 59.1±9.6 | 44.3±31. | 59.4±7.8 | 60.7±4.1 | 59.3±8.9 |
| 2 | 65.0±1.4 | 57.8±8.4 | 33.3±24. | 48.6±8.2 | 57.3±10. | 62.7±9.9 |
| 3 | 69.7±2.9 | 51.1±7.1 | 55.0±1.4 | 45.3±10. | 63.0±0.0 | 59.7±9.4 |
| 4 | 67.3±1.9 | 61.2±9.3 | 55.3±4.1 | 51.5±9.7 | 55.0±1.4 | 53.4±8.1 |
| 5 | 58.3±5.6 | 54.5±7.6 | 55.3±4.1 | 56.6±10. | 54.3±11. | 54.3±9.5 |
| 6 | 66.3±2.9 | 55.1±9.0 | 50.7±5.6 | 57.2±8.1 | 54.0±13. | 56.4±8.3 |
| 7 | 68.7±4.2 | 52.6±9.1 | 55.3±12. | 54.3±7.5 | 59.7±2.9 | 56.3±8.6 |
| 8 | 62.7±4.7 | 57.1±9.6 | 51.7±10. | 53.9±6.6 | 58.3±3.3 | 53.8±8.0 |
| 9 | 66.3±2.9 | 44.1±8.3 | 34.0±24. | 49.2±9.6 | 64.0±7.0 | 52.3±9.4 |
| 10 | 64.3±4.2 | 53.1±7.8 | 60.7±6.6 | 49.3±9.7 | 54.0±5.9 | 54.4±9.8 |
| 11 | 71.0±1.4 | 58.9±9.4 | 58.7±8.4 | 55.7±9.1 | 64.0±5.9 | 60.6±9.5 |
| 12 | 67.7±3.3 | 55.7±8.9 | 57.3±1.9 | 49.9±8.7 | 66.0±8.2 | 54.7±9.9 |
| 13 | 71.0±1.4 | 58.0±8.5 | 49.7±5.3 | 54.6±8.3 | 55.3±4.1 | 62.7±8.7 |
| 14 | 69.7±2.9 | 52.3±9.4 | 57.3±10. | 49.6±10. | 57.7±5.6 | 52.2±8.5 |
| 15 | 69.7±2.9 | 58.0±10. | 31.0±24. | 56.2±8.3 | 61.7±12. | 56.0±8.1 |
| 16 | 67.3±1.9 | 52.5±10. | 47.3±6.1 | 51.0±7.0 | 58.7±4.2 | 51.7±9.9 |
| 17 | 66.0±0.0 | 52.1±9.1 | 48.7±6.1 | 53.2±8.0 | 55.0±7.0 | 62.7±8.7 |
| 18 | 69.3±4.7 | 47.6±7.6 | 35.3±29. | 60.7±10. | 57.3±4.2 | 58.3±9.3 |
| 19 | 64.0±1.4 | 51.3±8.5 | 56.0±8.2 | 54.9±9.4 | 56.7±4.7 | 51.8±9.5 |
| 20 | 75.3±3.3 | 50.3±7.1 | 52.0±13. | 56.5±8.6 | 53.0±5.7 | 64.0±10. |
| 21 | 72.0±7.0 | 54.2±9.1 | 60.7±3.3 | 53.5±9.3 | 59.7±7.4 | 56.3±8.9 |
| 22 | 67.3±6.1 | 58.4±9.8 | 48.7±11. | 52.3±9.4 | 48.7±4.2 | 51.4±8.3 |
| 23 | 62.0±2.8 | 58.6±8.6 | 59.7±7.4 | 51.1±7.2 | 61.0±1.4 | 56.2±8.5 |
| 24 | 69.7±2.9 | 63.9±9.2 | 58.7±1.9 | 51.8±7.8 | 61.7±8.0 | 59.2±9.6 |
| 25 | 72.0±2.8 | 65.1±9.1 | 62.0±2.8 | 55.4±8.2 | 59.7±5.3 | 54.9±8.8 |
| 26 | 74.0±5.9 | 58.5±8.4 | 62.0±5.9 | 55.1±9.6 | 66.3±7.4 | 59.8±9.1 |
| 27 | 72.0±1.4 | 54.9±8.5 | 57.3±1.9 | 47.3±8.2 | 64.3±4.2 | 53.7±7.8 |
| 28 | 64.3±6.1 | 61.5±8.5 | 48.3±5.6 | 56.3±7.3 | 50.7±12. | 55.3±8.5 |
| 29 | 66.3±2.9 | 53.9±8.9 | 53.0±5.7 | 50.6±8.9 | 58.3±5.6 | 55.1±9.4 |
| 30 | 64.0±5.9 | 58.4±9.0 | 58.7±4.2 | 56.8±9.4 | 54.0±8.3 | 59.4±9.7 |
| 31 | 68.7±1.9 | 51.0±7.8 | 51.7±11. | 58.1±8.5 | 63.0±7.3 | 59.9±8.0 |
| 32 | 64.0±2.8 | 55.7±9.5 | 55.0±8.3 | 51.9±10. | 58.7±4.2 | 53.3±10. |
| 33 | 69.7±4.7 | 49.4±7.2 | 47.7±5.6 | 55.2±8.3 | 64.3±4.2 | 58.4±8.9 |
| 34 | 69.7±7.4 | 57.0±9.4 | 56.3±5.3 | 55.2±9.4 | 59.3±9.4 | 55.4±9.4 |
| 35 | 69.7±2.9 | 53.2±8.8 | 64.3±4.2 | 54.6±7.5 | 58.7±1.9 | 53.6±9.7 |
| 36 | 68.3±3.3 | 52.6±8.9 | 54.3±10. | 53.0±8.9 | 60.7±8.8 | 63.4±9.1 |
| 37 | 66.3±2.9 | 56.8±9.0 | 55.0±7.0 | 55.0±8.0 | 58.7±4.2 | 60.7±9.0 |
| 38 | 68.7±4.2 | 52.8±9.5 | 55.3±7.5 | 56.2±10. | 56.3±2.9 | 51.7±8.9 |
| 39 | 67.3±4.2 | 56.2±7.7 | 57.7±13. | 57.7±8.6 | 55.0±7.0 | 49.8±10. |
| 40 | 69.7±2.9 | 53.7±10. | 62.0±2.8 | 54.6±9.7 | 57.3±8.4 | 60.4±10. |
| 41 | 68.3±5.6 | 53.3±7.5 | 60.7±10. | 59.8±9.2 | 65.3±8.8 | 55.4±8.2 |
| 42 | 68.7±1.9 | 58.9±9.5 | 62.0±5.9 | 57.3±10. | 58.3±3.3 | 63.6±9.2 |
| 43 | 74.0±7.0 | 52.5±9.2 | 60.7±8.8 | 57.3±9.8 | 78.7±6.1 | 63.3±9.0 |
| 44 | 68.3±3.3 | 57.4±7.8 | 62.0±1.4 | 57.1±9.0 | 50.7±4.1 | 54.4±7.8 |
| 45 | 64.0±1.4 | 55.4±9.4 | 55.3±4.1 | 50.9±10. | 51.7±6.1 | 62.8±8.7 |
| 46 | 68.3±3.3 | 53.7±8.5 | 58.7±6.6 | 62.6±9.1 | 57.3±6.6 | 56.2±8.4 |
| 47 | 65.0±1.4 | 60.2±10. | 31.0±22. | 47.0±9.7 | 40.7±5.6 | 56.5±9.3 |
| 48 | 65.0±1.4 | 50.9±8.9 | 57.7±3.3 | 46.0±8.3 | 58.3±8.8 | 54.9±8.8 |
| 49 | 68.7±4.2 | 57.7±10. | 54.3±4.2 | 55.3±8.9 | 53.0±12. | 50.3±7.6 |
| 50 | 70.7±3.3 | 53.6±9.8 | 49.7±2.9 | 52.7±9.1 | 50.7±4.1 | 55.4±8.4 |
| 51 | 67.3±1.9 | 57.8±9.0 | 54.3±6.1 | 56.3±8.2 | 57.3±12. | 54.8±9.4 |
| 52 | 66.7±4.7 | 51.9±10. | 68.3±3.3 | 59.9±8.5 | 54.0±2.8 | 52.9±7.6 |
| 53 | 62.0±5.9 | 60.5±8.8 | 54.0±8.3 | 48.4±8.5 | 43.0±15. | 54.9±9.6 |
| 54 | 74.0±5.9 | 58.5±8.4 | 62.0±5.9 | 55.1±9.6 | 66.3±7.4 | 59.8±9.1 |
| 55 | 68.7±4.2 | 52.6±9.1 | 55.3±12. | 54.3±7.5 | 59.7±2.9 | 56.3±8.6 |
| 56 | 72.0±5.9 | 58.4±10. | 56.3±2.9 | 54.8±9.3 | 56.7±9.4 | 55.7±8.3 |
| 57 | 67.7±8.8 | 57.0±8.6 | 53.0±12. | 52.0±8.5 | 59.7±4.7 | 56.2±8.5 |
| 58 | 72.0±7.0 | 61.3±8.6 | 59.7±2.9 | 54.0±9.7 | 68.7±10. | 60.9±9.4 |
| 59 | 67.7±3.3 | 55.6±8.6 | 58.7±6.6 | 48.5±8.6 | 51.7±8.0 | 56.2±8.5 |
| 60 | 66.3±7.4 | 56.1±8.6 | 63.0±7.3 | 46.1±9.5 | 47.3±4.2 | 54.1±8.6 |
| 61 | 71.7±4.2 | 48.7±7.8 | 51.0±1.4 | 57.0±10. | 58.7±4.2 | 59.4±9.3 |
| 62 | 66.7±4.7 | 59.7±10. | 39.7±28. | 54.6±9.5 | 64.3±6.1 | 62.5±8.6 |

Table 5: Performance of each model for group B (Accuracy and Standard deviation: %)

| Subject ID | Our mothods | | Excluding reconstruction loss | | Excluding triplet loss | |
|---|---|---|---|---|---|---|
| | SP | NSP | SP | NSP | SP | NSP |
| 63 | 72.0±8.6 | 63.0±8.4 | 54.0±5.9 | 53.9±8.9 | 62.0±8.3 | 58.4±8.9 |
| 64 | 68.3±5.6 | 59.1±8.5 | 59.7±11. | 52.7±8.2 | 64.0±2.8 | 62.8±9.7 |
| 65 | 71.0±1.4 | 57.5±7.6 | 65.3±4.1 | 58.4±7.9 | 63.0±5.7 | 55.3±9.0 |
| 66 | 67.7±8.8 | 57.0±9.0 | 54.0±2.8 | 42.8±8.6 | 61.0±8.3 | 60.1±8.9 |
| 67 | 68.7±4.2 | 63.9±8.9 | 51.7±4.2 | 54.7±8.6 | 65.3±8.8 | 60.6±8.2 |
| 68 | 71.7±6.1 | 60.3±9.3 | 56.7±4.7 | 59.7±8.4 | 67.3±4.2 | 62.2±9.2 |
| 69 | 68.7±4.2 | 57.1±8.6 | 55.0±7.0 | 51.6±8.4 | 65.0±7.0 | 53.6±9.5 |
| 70 | 68.7±4.2 | 52.1±7.9 | 56.3±9.4 | 58.2±10. | 64.3±4.2 | 52.9±8.2 |
| 71 | 68.7±4.2 | 58.4±9.1 | 49.7±10. | 59.1±9.0 | 62.7±4.7 | 55.9±10. |
| 72 | 68.7±6.6 | 54.7±8.2 | 55.3±7.5 | 48.2±10. | 50.7±14. | 60.0±7.5 |
| 73 | 66.3±2.9 | 54.9±7.1 | 63.3±4.7 | 62.2±8.5 | 61.7±4.2 | 60.9±9.2 |
| 74 | 73.0±2.4 | 60.6±8.0 | 49.7±5.3 | 54.2±8.6 | 63.0±9.6 | 62.5±9.3 |
| 75 | 65.3±3.3 | 59.4±6.9 | 70.0±8.2 | 59.8±4.5 | 62.7±4.7 | 61.9±9.5 |
| 76 | 67.3±6.6 | 58.6±9.1 | 56.3±4.7 | 52.6±7.4 | 58.7±4.2 | 54.4±9.0 |
| 77 | 70.7±4.1 | 64.8±8.0 | 60.7±7.5 | 58.9±10. | 64.3±4.2 | 57.7±7.9 |
| 78 | 65.0±1.4 | 59.9±8.5 | 53.0±7.3 | 47.2±8.3 | 56.3±7.4 | 61.1±8.1 |
| 79 | 67.7±3.3 | 57.5±7.6 | 51.0±15. | 62.2±8.6 | 58.3±5.6 | 62.0±9.0 |
| 80 | 68.7±1.9 | 56.8±9.0 | 57.7±11. | 61.7±8.5 | 65.3±4.1 | 59.1±8.5 |
| 81 | 67.7±5.6 | 58.4±8.7 | 58.7±8.4 | 56.9±10. | 60.0±14. | 61.0±8.0 |
| 82 | 69.3±4.7 | 56.1±8.5 | 48.7±9.8 | 58.6±8.6 | 58.7±6.1 | 61.6±9.9 |
| 83 | 65.0±1.4 | 48.9±7.8 | 55.0±9.4 | 58.7±8.2 | 59.7±2.9 | 61.0±9.2 |
| 84 | 71.0±1.4 | 60.5±8.1 | 56.3±7.4 | 52.1±9.7 | 60.7±6.6 | 60.5±8.2 |
| 85 | 67.3±1.9 | 55.9±7.8 | 52.7±4.7 | 47.8±8.4 | 59.3±4.7 | 61.5±9.8 |
| 86 | 64.0±1.4 | 60.6±8.1 | 53.0±5.7 | 52.1±8.1 | 64.3±6.1 | 56.4±9.0 |
| 87 | 68.3±5.6 | 55.9±8.6 | 59.7±5.3 | 60.7±9.3 | 58.7±9.8 | 63.2±8.9 |
| 88 | 64.0±1.4 | 56.9±10. | 58.7±9.8 | 59.9±8.0 | 49.3±4.7 | 54.7±8.8 |
| 89 | 66.3±5.3 | 62.0±9.2 | 48.7±11. | 50.5±9.1 | 53.0±13. | 62.6±9.4 |
| 90 | 69.7±2.9 | 49.1±8.4 | 56.3±5.3 | 52.6±8.4 | 59.7±2.9 | 59.0±10. |
| 91 | 70.7±3.3 | 59.4±8.6 | 49.3±4.7 | 54.8±9.8 | 65.0±9.4 | 62.8±7.8 |
| 92 | 67.3±4.2 | 52.1±9.4 | 54.0±16. | 60.1±8.3 | 61.7±4.2 | 57.5±8.9 |
| 93 | 73.0±5.7 | 55.7±7.8 | 57.7±3.3 | 42.7±8.7 | 65.3±11. | 53.5±9.3 |
| 94 | 74.0±7.0 | 61.5±8.2 | 66.3±2.9 | 59.8±7.9 | 67.7±5.6 | 63.1±8.8 |
| 95 | 70.7±4.1 | 58.2±8.5 | 44.0±8.6 | 59.0±9.0 | 72.0±2.8 | 63.6±8.0 |
| 96 | 68.7±1.9 | 58.3±7.1 | 63.3±4.7 | 51.3±8.7 | 66.3±4.7 | 60.5±9.5 |
| 97 | 66.3±2.9 | 50.9±8.5 | 56.3±4.7 | 58.9±9.9 | 65.3±4.1 | 62.2±7.8 |
| 98 | 68.3±5.6 | 60.7±10. | 49.7±2.9 | 51.7±8.9 | 67.7±10. | 62.1±8.9 |
| 99 | 65.3±8.8 | 57.9±8.1 | 60.7±7.5 | 50.6±9.7 | 56.7±9.4 | 59.9±8.9 |
| 100 | 67.3±1.9 | 59.7±9.4 | 53.0±2.4 | 50.6±7.8 | 67.3±8.4 | 61.0±9.4 |
| 101 | 73.0±5.7 | 58.6±8.4 | 48.7±9.8 | 57.6±8.6 | 63.3±4.7 | 61.0±8.8 |
| 102 | 73.3±4.7 | 57.8±9.2 | 58.3±10. | 55.8±10. | 62.0±5.9 | 62.0±9.0 |
| 103 | 73.0±2.4 | 61.8±8.2 | 50.0±8.2 | 54.1±8.7 | 68.3±3.3 | 52.6±9.2 |
| 104 | 69.7±9.7 | 56.7±8.8 | 55.0±9.4 | 57.2±10. | 75.0±1.4 | 61.7±9.5 |
| 105 | 66.3±5.3 | 62.5±9.4 | 54.0±8.6 | 59.1±9.4 | 66.0±14. | 62.1±9.8 |
| 106 | 74.0±1.4 | 52.2±9.4 | 64.3±8.0 | 57.4±7.7 | 64.0±13. | 62.9±9.2 |
| 107 | 71.0±1.4 | 58.3±10. | 56.3±4.7 | 56.7±7.3 | 66.3±4.7 | 62.7±9.5 |
| 108 | 66.3±2.9 | 48.8±9.1 | 49.7±10. | 52.1±7.6 | 65.3±4.1 | 57.8±7.8 |
| 109 | 68.7±1.9 | 55.5±9.1 | 55.3±5.6 | 50.5±8.2 | 58.7±9.8 | 55.4±8.2 |
| 110 | 68.7±4.2 | 56.7±8.5 | 56.3±9.4 | 55.8±8.4 | 54.3±10. | 60.5±8.4 |
| 111 | 68.3±3.3 | 59.4±8.2 | 59.7±7.4 | 49.8±7.7 | 65.3±5.6 | 60.2±9.6 |
| 112 | 70.7±4.1 | 52.4±7.9 | 56.7±9.4 | 57.2±7.7 | 54.3±11. | 60.0±7.5 |
| 113 | 70.7±3.3 | 52.4±9.5 | 58.3±8.8 | 49.4±8.9 | 64.0±2.8 | 62.8±9.7 |
| 114 | 72.0±1.4 | 65.4±7.7 | 50.7±10. | 45.5±8.5 | 59.7±5.3 | 58.5±8.1 |
| 115 | 68.7±1.9 | 55.9±8.9 | 62.0±9.4 | 55.6±9.7 | 60.7±5.6 | 65.5±8.5 |
| 116 | 66.3±2.9 | 55.3±8.3 | 55.3±13. | 61.2±10. | 60.7±3.3 | 53.8±9.8 |
| 117 | 64.3±8.0 | 56.3±7.6 | 57.3±9.8 | 62.6±8.8 | 60.7±8.8 | 60.3±8.2 |
| 118 | 71.7±4.2 | 54.4±7.7 | 50.7±13. | 54.9±8.9 | 63.0±12. | 61.2±8.8 |
| 119 | 67.7±3.3 | 50.9±8.5 | 52.0±15. | 59.2±8.7 | 58.3±8.8 | 59.3±8.6 |
| 120 | 69.7±2.9 | 52.2±9.2 | 59.7±7.4 | 56.3±8.5 | 58.7±1.9 | 62.1±9.0 |
| 121 | 61.7±4.2 | 58.7±9.3 | 56.3±2.9 | 55.0±7.7 | 71.7±4.2 | 53.3±9.3 |

## A.6 PSEUDO-CODE FOR QUALITATIVE ANALYSIS

This pseudo-code summarizes the algorithm for selecting annotation pixels of the model in Section 4.6 of the main text.

---

**Algorithm 1** Identify key pixels based on embedding variance

---

1: **Input:** Embedding function $E(\cdot)$, Separated reference dataset $D$, Test image $e$, Number of repetitions $N$
2: **Output:** Top 10 key pixels
3:
4: **Step 1: Compute variances for each embedding Dimension**
5: **for** each image $d \in D$ **do**
6:     Compute embedding $E(d)$
7: **end for**
8: Compute variance for each embedding dimension
9: Determine reference unit as the dimension with the highest variance
10:
11: **Step 2: Analyze each pixel in test image** $e$
12: **for** each pixel $(i, j)$ in $e$ **do**
13:     $L$ : Empty list
14:     **for** $k = 1$ to $N$ **do**
15:         $E(e_{noisy}$ : Add random noise to pixel $(i, j)$ in $e$
16:         Compute embedding $E(e_{noisy}$
17:         Append reference unit output to $L$
18:     **end for**
19:     Compute variance of the $L$
20: **end for**
21:
22: **Step 3: Identify top 10 key pixels**
23: Select top 10 pixels with the highest variance in $L$
24: **Return** Top 10 key pixels

---

## A.7 HYPERPARAMETER SENSITIVITY ANALYSIS

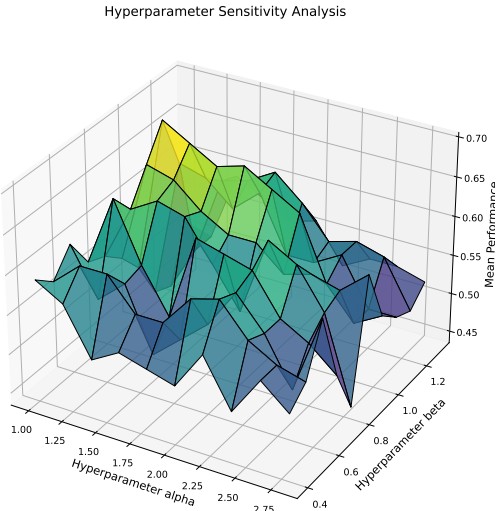

Figure 12: Sensitivity Analysis of Average Predictive Performance for Hyperparameters $\alpha$ and $\beta$.

The hyperparameters $\alpha$ and $\beta$, introduced in Section 4.4 of the main text, were determined through a preliminary sensitivity analysis using data from five subjects. $\alpha$ values were varied discretely between 1.0 and 2.0, and $\beta$ values between 0.4 and 1.3, generating a total of 100 parameter combinations for testing. Refer to Fig. 12. As a result, the highest average predictive performance of 0.7 was observed at $\alpha = 1.2$ and $\beta = 1.0$, which were selected as the optimal hyperparameters.

## A.8 TEST-RETEST ANALYSIS OF SIMILARITY PATTERN VECTORS

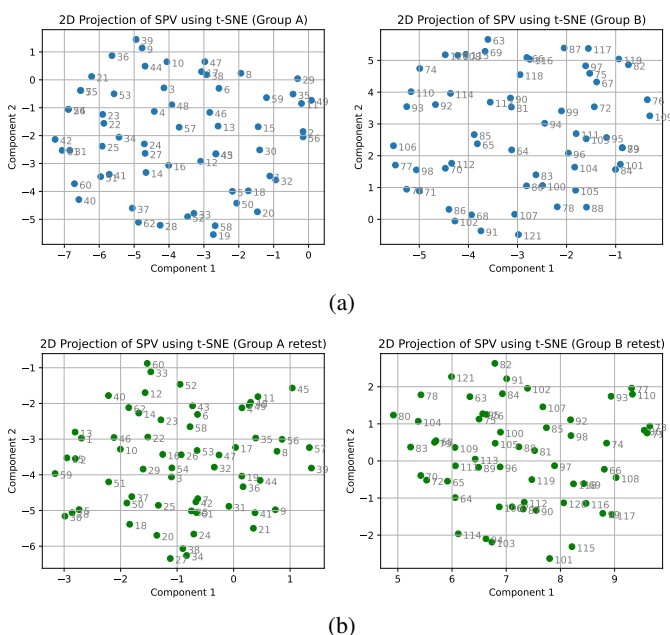

(a)

(b)

Figure 13: Results of the test-retest analysis for similarity pattern analysis across all subjects with 15 triplets. (a) 2D t-SNE visualization of similarity pattern vectors obtained from the initial sampling. (b) 2D t-SNE visualization of similarity pattern vectors obtained from the delayed sampling.

The SPVs (Similarity Pattern Vectors) of each subject were measured through behavioral experiments. To validate the reliability of similarity pattern sampling, we compared the SPVs obtained from re-sampling conducted at different time intervals. Behavioral sampling was performed for 1,500 triplets per subject, with 15 triplets randomly re-sampled during the experiment (Retest) to assess subject consistency. The SPVs generated for each subject from two samplings of the same 15 triplets were then compared. While some subjects exhibited varying responses to the same triplets, their SPVs were observed to cluster closely with those of similar subjects during the Retest. This finding suggests that subjects demonstrated high response consistency even when re-sampling was conducted after a temporal delay.

