# OpenReview forum: "Embedding Learning for Approximating Person-specific Cognitive Similarity"
_ICLR.cc/2025/Conference — Submitted to ICLR 2025_

### Official Review · Reviewer_tueP · 2024-10-25

**Soundness:** 2
**Presentation:** 1
**Contribution:** 2
**Rating:** 3
**Confidence:** 5

**Summary:**

The paper presents an approach for learning a human-like representational space using a deep learning architecture. This embedding space is supervised by human similarity judgments, specifically, for triplets of chest X-ray images.  The deep learning architecture is an autoencoder, and the representation learning is implemented using a triplet loss function, whose purpose is to produce an embedding space where objects (images) perceived as more similar by humans are positioned closer together. The model successfully generalizes to predict human similarity judgments.

**Strengths:**

The main contribution of this paper is its demonstration that it is possible to learn person-specific embeddings, as seen in good out-of-sample prediction of similarity relations at the individual level.  Additionally, the findings suggest that person-specific embeddings effectively capture unique, idiosyncratic aspects of each person's similarity judgments. This is seen in the fact that models learned for one participant do not perform well in predicting similarity judgments of other participants.

**Weaknesses:**

1) Novelty of embedding workflow: The novelty of the paper is somewhat limited, because several studies have already presented related methods for learning embedding spaces that align object distances with human similarity judgments, or related tasks like the triplet task (where a person selects which of two objects is more similar to a reference object). Palazzo et al. (2020) use a triplet loss to learn a joint embedding of EEG data and visual images; Zhang et al. (2018) present LPIPS which reweights layers of a pre-existing network based on human triplet judgments;  Tarigopula et al. (2023) use pruning to improve the alignment between object-distances in the embedding space and human similarity judgments;  Jha et al. (2023) extract low-dimensional representations from pre-trained CNNs using similarity learning in a lower-rank embedding space so that distances in the embedding space maintain monotonic relation with human similarity judgments.  Given these prior studies, the main differences in the present study are the use of an autoencoder and learning/modeling single-participant data. However, even these differences are associated with some weaknesses as detailed below.

2) Some architecture and implementation details are not well explained or justified. In particular, it is not clear why the authors choose to use an autoencoder architecture rather than a simpler encoder operating on features produced from a separate feature extractor.  In principle, it should be possible to learn an embedding space without the decoder (see Jha et al, Palazzo et al.). That would produce a smaller, less complex model.  Looking at the details of the ablation study it is clear that the addition of the decoder improves performance, as compared to using the triplet loss alone, and the reason is probably that the decoder is necessary for learning a rich set of visual features.  An alternative (used by prior studies) is to use a pretrained CNN as a feature extractor, and then rely only on an encoder alone for metric learning.  A potential weakness in using the autoencoder is that the object-distances end up relying on features that are also required to reconstructing the image, but it is not clear whether those are necessarily the most important drivers of human similarity judgments.  Another reason for using a pre-trained network as a feature extractor is suggested from the ablation exercise presented in Table 1. It shows that using a reconstruction (decoding) loss alone produces above-chance performance in predicting human judgments. This suggests that certain image features, captured by the decoder, drive similarity judgments across the group. This strengthens the argument for using a strong pretrained architecture as feature extractor, which could then be fine-tuned on subject-specific data.

3) Absence of justification for modeling individual data: The abstract and introduction write that “modeling psychological embeddings at the individual level can be beneficial,” but the authors do not provide a clear demonstration of how single-participant modeling improves a specific objective or task. For certain applications, such as identifying individuals who may differ from others, it is not even necessary to use an embedding space; such analyses can be conducted directly from the similarity matrices computed from the behavioral data.

4) Novelty and strength of interpretability analysis (section 4.6): the authors introduce an interpretability analysis whose purpose is to identify important parts of the images. It has a few weaknesses. First, the details are not presented in a separate methods section but introduced on the fly in the results. Second, the analysis choices are not argued for. The indicator of quality to be explained is the variance in a single reference unit of the network, which presents the strongest variance for a batch of images. They then define more important image pixels as those whose masking reduces the variance in this unit.  Both Palazzo et al. and Tarigopula et al. report related masking procedures, but in those studies, the impact of masking was evaluated by determining how the masking of each pixel (or image region) impacts the alignment between the DNN and human similarity spaces, which is a more direct test of which image areas are psychologically relevant than the test evaluated here.  As a consequence, the novelty and validity of the masking procedure suggested here is weakened.
Separately from this issue, a formal quantitative overlap between human raters and the interpretability measure is missing; only a qualitative evaluation is provided via a figure.

**Questions:**

it was not clear to me why the loss term called ‘variable triplet loss’ was used. The traditional triplet loss forces a solution where D(a,c) < [D(a,d)+margin] where D is distance, a the anchor and c,d the similar and dissimilar objects.  To my understanding, the loss term used (stronger weight on distance to closer anchor) will encourage D(a,c) < D(a,d) but does not force it. That is, there could be solutions where this does not hold. The choice of this term should be better motivated.

Additional feedback and references mentioned
Re: Figure 3 – The figure shows how participants are positioned in a lower-dimensional space. To interpret these distances, it would be good to include a test-retest measure for each participant, which would formally quantify intra-participant variability, not just inter-participant variability.
p. 1 Re’ the statement that “the amount of person-specific behavioral data that can be collected through similarity behavior sampling is insufficient in most scenarios”, and similar statements in Section 2.1: There are effective multi-item arrangement methods, similar to the procedure used here that allow estimating object similarity in multiple dimensions (Kriegeskorte and Mur, 2012).
p.2 Re’ the statement “we conducted a first-ever behavioral sampling experiment to measure the cognitive similarity of actual CXR images with 121 clinical physicians, focusing on realistic scenarios.” This seems to be an important point, but it was not clear what does the similarity of medical images measure? If these judgments are independent of diagnosis (as appears to be the case here), the dimensions that drive similarity might be completely unconstrained and left to each person’s own interpretation. This means that it’s possible that two physicians could make very different similarity judgments even if they arrive at the same diagnosis. It would be interesting to know whether these similarity judgments correlate with agreement on diagnosis.
P. 4 The authors mention a limitation of person-specific modeling, writing, “behavioral data from a subject can typically only be used to train an individual model for that subject.” The word ‘only’ was unclear; the judgments could be averaged  to create a group -level similarity matrix if the triplets are same across participants.
p. 5 It’s not completely clear how the binary labeling was applied to a triplet so that concatenation produced SPV.
p. 6 The participants were 121 clinical physicians.  They seem to vary widely over age/experience (Appendix; Table 2 Min = 26; Max = 55 years of age). It’s probable they differ considerably in their ability to evaluate chest X-ray images. It could be interesting to see whether the embeddings or behavior are more similar among the more experienced participants.
p. 7 The method for applying t-SNE to binary strings of SPV is unclear. Binary data require specific distance functions, and those details are missing here.
p. 7 section 4.5 clunky writing around the text in parentheses.

Refs
Palazzo, S., Spampinato, C., Kavasidis, I., Giordano, D., Schmidt, J., & Shah, M. (2020). Decoding brain representations by multimodal learning of neural activity and visual features. IEEE Transactions on Pattern Analysis and Machine Intelligence, 43(11), 3833-3849.
Zhang, R., Isola, P., Efros, A. A., Shechtman, E., & Wang, O. (2018). The unreasonable effectiveness of deep features as a perceptual metric. In Proceedings of the IEEE conference on computer vision and pattern recognition (pp. 586-595).
Jha, A., Peterson, J. C., & Griffiths, T. L. (2023). Extracting low‐dimensional psychological representations from convolutional neural networks. Cognitive science, 47(1), e13226.
Tarigopula, P., Fairhall, S. L., Bavaresco, A., Truong, N., & Hasson, U. (2023). Improved prediction of behavioral and neural similarity spaces using pruned DNNs. Neural Networks, 168, 89-104.
Kriegeskorte, N., & Mur, M. (2012). Inverse MDS: Inferring dissimilarity structure from multiple item arrangements. Frontiers in psychology, 3, 245.

---

> ### Author Response · Authors · 2024-11-18
> **We deeply appreciate the thorough review comments (Response 1)**
>
> We deeply appreciate that the reviewer has thoroughly understood and carefully reviewed the details of our manuscript, providing invaluable feedback to improve our work. Personally, I have learned a lot from your comments.
> I hope the following response addresses your concerns.
>
>
> 1. Novelty of the embedding workflow
>
>
> Thank you for providing a detailed account of prior related works that we may have overlooked. We will cited all the studies you mentioned in our paper and will take them into careful consideration for our future work. We would like to emphasize again that the novelty of our study lies in modeling expert data with high complexity. Most existing studies use benchmark datasets or data with lower complexity, where inter-human similarity perception tends to show minimal variability. To the best of our knowledge, no prior work has provided evidence that the patterns of similarity perception differ significantly among individuals when using the datasets employed in those studies.
>
> In contrast, we specifically selected datasets where similarity perception is more likely to vary across individuals, and we provided evidence supporting this claim (Sec 4.3). Moreover, conducting behavioral experiments with experts is inherently challenging and rare, and we believe that this independent contribution should be acknowledged to encourage further work despite the high costs and risks associated with such studies. The expert behavioral data we are making publicly available through ICLR will, we believe, serve as a valuable resource for researchers in this field.
>
>
> 2.	Rationale for Using Autoencoders Instead of Encoders
>
> We agree with the reviewer’s comment that, considering the properties of CNNs, an encoder alone can extract substantial visual cognitive features. However, in general, training an encoder requires labeled data, which could constrain the type of features it learns. Since humans, especially in the case of expert data, do not necessarily judge the similarity of data based solely on labels, it seems difficult to definitively conclude that features learned solely by an encoder are a better alternative than an autoencoder for person-specific embedding learning.
>
> While autoencoders have limitations in learning based on the features necessary for image reconstruction, their label independence suggests that they may uncover diverse manifolds that could provide a better understanding of image similarities. We are not claiming that the autoencoder is the optimal architecture for person-specific embedding learning. Rather, we wish to emphasize that unsupervised learning, independent of labels, when combined with triplet loss, has shown the potential to amplify limited individual human behavior sampling information.
>
> We apologize if our explanation did not adhere to strict mathematical definitions. We will make an effort to include comparative results with encoder-only models in the revised version of the manuscript, which we plan to upload within the next few days.
>
>
> 3. Justification for modeling individual data
>
> We would like to emphasize that the ultimate goal of this work goes beyond simply identifying different individuals at the behavioral level. For example, experts may possess vastly different knowledge and skills. If there is an expert with a very high level of expertise, we could potentially gain inspiration for implementing high-performance machine learning models by reconstructing the expert's embedding. Alternatively, for the individualized, customized learning of expertise, it may crucial to identify which data points hold high uncertainty for an individual. Since the uncertainty of highly similar data tends to be similar, if we can reconstruct an individual's embedding, we could refine the individual's learning through personalized uncertainty estimation.
>
> 4. Interpretability analysis
>
> We apologize for any confusion caused. In the revised version, we will move the details to the methods section and quantify the part that was previously only qualitatively assessed through visuals, presenting it in a table format. As the reviewer pointed out, the novelty of the masking procedure cannot be considered as part of our contribution. However, as intended, the masking experiment serves as an auxiliary analysis to demonstrate the validity of our methodology, and we kindly ask that you consider this in the context of our intention not to emphasize the novelty of the masking methods themselves.

---

> ### Author Response · Authors · 2024-11-18
> **We deeply appreciate the thorough review comments (Response 2)**
>
> 5. Triplet Loss
>
> As the reviewer pointed out, traditional triplet loss enforces a margin with the condition D(a,c) < [D(a,d) + margin]. However, our triplet loss does not explicitly consider a margin. We would like to clarify that this choice is intentional. Unlike traditional metric learning, which learns an average metric for the general population, the person-specific similarity sampling data for expert data is limited and can reflect considerable uncertainty. For instance, when sampling similarity from 100 sets, 5-10 of those samples may deviate from the true similarity trend of the subject. Instead of forcing the learning of similarity from noisy data that arises from human behavior sampling via a margin, our goal is to prevent the similarity of data that deviates from the trend from being incorporated into the model.
>
>
> 6. Figure 3 (test-retest analysis)
>
> Behavior sampling for each subject was conducted 500 times, divided across intervals of more than one day. While it is challenging to perform a rigorous test-retest analysis under the given constraints, we will reanalyze the divided datasets collected at different time intervals. The observation of similar trends in the two behavior datasets sampled with time intervals will be incorporated into the appendix materials of the revised version.
>
>
> 7. Relationship between diagnosis and similarity
>
> We agree with the reviewer’s opinion and find this to be a very intriguing topic. In fact, we conducted separate medical imaging diagnostic tests for all subjects apart from the similarity measurement experiments. However, we refrained from analyzing the diagnostic results, as we were concerned that it might dilute the focus of this study on embedding modeling. Nevertheless, we have accepted the reviewer’s suggestion and performed a brief analysis of the relationship between diagnostic ability and similarity perception patterns. This analysis will be incorporated into the revised edition, which will be uploaded in a few days.
>
> To summarize the results:
>
>
> - In the CXR-A group, the similarity pattern vectors of physicians with superior diagnostic abilities tended to cluster closer together, but no distinct clusters were formed.
> - In the CXR-B group, however, the similarity pattern vectors of highly skilled diagnosticians showed a tendency to form distinct clusters.
> This supports our hypothesis that CXR-B involves a higher tendency for similarity perception centered around active lesions.
>
>
> We believe this topic could be an extremely important independent research subject. Therefore, we plan to address this as an independent focus in future work, reflecting the reviewer’s excellent feedback.
>
>
> Additionally, while diagnostic ability varies by a physician’s age or experience, we do not believe that diagnostic ability simply correlates with age or years of clinical experience. Rather, diagnostic ability is likely proportional to the amount of time actively spent performing diagnostic tasks. Therefore, a detailed investigation into precise clinical experience may be necessary.
>
>
>
> 8. Binary Strings of SPV
>
>
> The binary strings of SPV (Similarity Pattern Vector) function as a type of one-hot vector, where each similarity measurement task is substituted as a single dimension. For example, if there is one anchor and two comparison images, there are two possible similarity outcomes, which can be represented as either 0 or 1. If there are 500 such task sets, a 500-dimensional SPV can be defined.
>
> While this one-hot vector is very simple, it lacks weights between dimensions, meaning it can straightforwardly represent the similarity of similarity patterns (a form of meta-similarity, as we conceptualize it) using basic Euclidean distances. For the same reason, reducing the dimensionality of this one-hot vector and visualizing it with t-SNE does not pose any technical flaws.
>
>
> 9. Similar statements in Section 2.1 and other presentation issues
>
> Thank you for the suggestion. The similar statements in Section 2.1, awkward wording in Section 4.5, and issues with the use of "only" will be revised and reflected in the updated version.

---

### Official Review · Reviewer_yX4C · 2024-11-04

**Soundness:** 3
**Presentation:** 3
**Contribution:** 2
**Rating:** 6
**Confidence:** 4

**Summary:**

This paper presents an investigation of using a small amount of similarity sampling data to fine-tune pretrained embeddings and learn person-specific embeddings.  The authors evaluate this on a medical imaging task and perform an impressive scale evaluation with 121 clinical physicians.  Person specific/ personalization analyses are interesting and valuable as there many tasks in which there are individual differences and performance can be improved significantly by adapting to a user.

**Strengths:**

The paper has several strengths, it is well written and clearly presented overall.  The data collection is impressive and valuable to the research community.  The paper generally has sufficient details for replication.

**Weaknesses:**

The authors claim theirs is  “first large-scale experimental study to model individual-level psychological embeddings” This is a big claim and I don’t think it is really justified. There are many papers on personalization and subjective tasks such as emotion labeling in affective computing.  I would request the authors to clarify what the novelty is or if I have misunderstood the approach.
Related to this I think the paper should have a more extensive related work section on personalization that helps the reader understand the differences between this work and other adaptive/personalized models.

The details of the data collection are a little unclear.  1) How many images did each clinician label?  Did they all complete 500 exactly? What was the range of times it took for them to complete these?  This is a might seem a little like I am nitpicking, but the authors claim the data collection as one of their three main contributions and so it would be great to have a little more detail about the data collection over all, including (a) the background of the clinicians, (b) the average and range of number of years of experience, (c) more details about the instructions they were given.

The data collection is impressive and the study is interesting.  I commend the authors on this.  The results also support the performance claims showing a consistent bump in accuracy.  This is not particularly surprising given that personalization usually leads to better results than a generic one.
Will the data be released?  I apologies if I missed something but again the value of these data for future research could be significant.

I did not quite understand the message behind Fig.4 (b) - Are they both highlighting that there is little correlation between the two?

In the introduction key contributions numbered points the text in parentheses e.g. “Perspectives on cognitive science” seem unnecessary.  I would remove these.

Overall, the paper is well written and motivates the work well.  I think this has potential, however, it would be helpful if the work is positioned more clearly in the literature and the contributions contextualized within that.

**Questions:**

See above.

---

> ### Author Response · Authors · 2024-11-18
> **Thank you for your constructive comments (Response 1)**
>
> Thank you for your constructive comments. I sincerely appreciate your acknowledgment of the potential of our work and your recognition of the value of the data. I hope the responses below address your concerns effectively.
>
>
> 1. Novelty of our work compared to previous study
>
> We have carefully reviewed the reviewer’s comments and acknowledge the possibility that our claims may be overly assertive. We will reflect this in the revised version, which will be uploaded in the coming days.
> Separately, we would like to highlight several aspects of our study that distinguish it from previous studies:
>
> (1) Use of complex expert data for individual embedding modeling:
>
> Our work attempts to model individual embeddings of experts (non-radiologist physicians) using complex, practical expert data, namely medical imaging (chest X-rays). To the best of our knowledge, no studies have explored individual embedding modeling using real-world (expert) data. It is well established that the dimensionality required to determine similarity for general image data is relatively low (e.g., Jha, A., Peterson, J. C., & Griffiths, T. L. (2023). Extracting low‐dimensional psychological representations from convolutional neural networks. Cognitive Science, 47(1), e13226). For general datasets with clear labels, individuals can perceive similarities based on distinct attributes, resulting in minimal variation in similarity patterns among individuals. For example, when comparing images of a dog, a cat, and a snake, most people would consider the dog and cat to be more similar.
>
> In contrast, expert data involves significantly higher dimensionality in determining similarity. For instance, in the domain of chest X-rays, even if two physicians arrive at the same diagnosis, the pathways and patterns they use to reach that conclusion can vary greatly. Demonstrating that individual metric learning is feasible with such complex expert data and building evidence to support this capability represents a meaningful contribution to the metric learning community, separate from studies using general datasets.
>
>
> (2) Demonstrating the utility of unsupervised learning (Autoencoders) in person-specific metric learning:
>
> In addition to showing the feasibility of individual metric learning with expert data, our work highlights the novelty of addressing practical challenges in this domain through unsupervised learning (autoencoders). In domains with complex data, such as medical imaging, where individual similarity perception patterns vary, the amount of similarity information that can be sampled (measured) from a person is extremely limited.
>
> For instance, in our experiment, collecting 500 samples per individual required over 5 hours on average. Considering fatigue and the constraints of experts' working hours, continuous measurement is challenging, requiring significant time and effort overall. Despite this, the 500 samples collected are insufficient for robust modeling given the complexity of the image domain. Conventional modeling approaches struggle to achieve meaningful predictive performance with such limited data.
>
> To address this, we integrated unsupervised feature extraction through autoencoders with traditional methods. There are no prior examples in metric learning where unsupervised learning was combined to improve performance. Autoencoders do not require additional information to train but can flexibly learn the manifold structure in an unsupervised manner. (It can be explained that the autoencoder amplifies the individual similarity information obtained from a small number of samples) Our experiments demonstrated that this capability can be harnessed to extract characteristics specific to individual human learners, which we believe is a novel contribution.
>
> In response to the reviewer’s comments, we will revise the claim of being the "first large-scale experimental study" to limit it to the "expert domain" and strengthen the Related Works section in the revised version.
>
>
> 2. Details of data collection
>
> In each set, three images were presented, and clinical physician participants evaluated the similarity by comparing the three images. Each participant completed a total of 500 sets without exception. Participants who withdrew were excluded from the analysis. The total time spent, as clearly mentioned on lines 333-334 of this paper, was an average of 304 minutes for the CXR-A group and 245 minutes for the CXR-B group. Details of the clinical physicians' background, age, and other information related to the data collection process are provided in Appendix 1 at the end of the paper. Further detailed information about the participants' data is also provided in the supplementary file.

---

> ### Author Response · Authors · 2024-11-18
> **Thank you for your constructive comments (Response 2)**
>
> 3. Consistency of results / Data release
>
>
> We deeply appreciate your recognition of our potential and efforts. While it may seem obvious that personalized modeling yields better results than general metric modeling, we would like to emphasize once again that achieving meaningful personalized modeling outcomes is quite challenging, given the limited amount of behavioral data that can be obtained for each individual. What we find particularly remarkable is that, despite the lack of additional sampled information for each individual, the model enhanced with an autoencoder appears to amplify the limited individual data effectively.
>
> The data will, of course, be made available through ICLR 2025. We believe that collecting and sharing high-cost data, with expert subjects, is a significant contribution to the machine learning community. If this contribution is recognized, we hope it will encourage other institutions to collect and share similar high-cost data, ultimately fostering study that makes use of such relevant data.
>
>
>
> 4. Figure 4(b)
>
>
> We apologize for the insufficient explanation. In Fig. 4 (b), the diagonal represents the results of testing a specific subject’s model on that subject’s test data. These values correspond to the light blue (SP) bars in Fig. 4 (a). The blue bars (NSP) in Fig. 4 (a) represent the average of the results from testing a specific subject’s model on the test data of other subjects, excluding that particular subject. The values outside the diagonal in Fig. 4 (b) represent the results of for each test data point from other subjects, rather than the average.
> According to our definition, the average prediction performance on test data from other subjects is the NSP. Therefore, the average of all the values in each row of Fig. 4 (b), excluding the values above the diagonal, represents the NSP for each model.
> (Please let me know if my explanation is unclear or insufficient.)
>
> Note that the image sets used for modeling and the image sets used for performance testing of the trained models are the same for all subjects within the group. Since modeling is done independently for each individual, the model trained on a specific subject’s data should perform well on that subject’s test data but should show lower performance on test data from other subjects.
>
>
> 5. Unnecessary text
>
> I agree with the reviewer’s comment. We will incorporate the reviewer’s suggestions in the revised version to be uploaded in the next few days, removing or modifying the relevant text.

---

> > ### Comment · Reviewer_yX4C · 2024-11-25
> > **Response to Rebuttal**
> >
> > I would like to thank the authors for their rebuttal.   I do feel that their thoughtful comments address some of my comments on the paper.  I would be willing to consider changing my score; however, I cannot see a revised version of the paper uploaded and as such it is a little hard to evaluate the actual modifications that have been implemented.  If the authors have upload a revision can they let me know?  Ideally that revision would have marked up changes to make it easier to evaluate.

---

> > > ### Author Response · Authors · 2024-11-25
> > > **Thank you for your understanding**
> > >
> > > We are currently conducting additional analyses, which has delayed the upload of the revised version of the paper. We will notify you as soon as it is uploaded. We deeply appreciate your patience and consideration.

---

> > > ### Author Response · Authors · 2024-11-25
> > >
> > > Dear Reviewer,
> > >
> > > Thank you for your patience. The revised version of the paper has been updated.
> > > Please note that there may be additional updates before the final deadline, as we continue to incorporate further feedback and analysis.
> > >
> > > Regarding your concerns, the following changes have been made in the revised version:
> > >
> > > 1. Revised Section 2.2 - "PERSON-SPECIFIC COGNITIVE SIMILARITY MODELING":
> > >
> > > In response to your comment, we have extensively rewritten this section to cover a broader range of work related to personalization. We highlight that numerous feature engineering studies have been conducted for individual-level embedding modeling. However, we point out that these existing studies have not addressed practical issues such as the lack of individual-specific sampling in person-specific modeling. We emphasize that our approach, combining unsupervised feature learning, aims to resolve the issue of limited individual-specific data, which has been largely overlooked in previous research. Additionally, we clarify that previous embedding modeling studies primarily used benchmark datasets with minimal inter-individual variation in similarity metrics, whereas our study is novel in utilizing expert data with significant metric differences at the individual level.
> > >
> > > 2. Modification of Big Claim:
> > >
> > > The claim on Line 101 has been revised from "first large-scale experimental study to model individual-level psychological embeddings" to "first expert-based experimental study to model individual-level psychological embeddings" to provide more specific context.
> > >
> > > 3. Removal of Unnecessary Text:
> > >
> > >  We have removed unnecessary text from Lines 101-107.
> > >
> > > Please be aware that additional revisions may still be made.
> > >
> > > Kind regards,

---

> > > > ### Comment · Reviewer_yX4C · 2024-12-03
> > > > **Resoonse**
> > > >
> > > > I have updated my review and score in response to the updated manuscript. Thank you.

---

> > > > > ### Author Response · Authors · 2024-12-04
> > > > >
> > > > > We would like to express our sincere gratitude once again for the constructive comments from the reviewer and the reevaluation of our paper.

---

### Official Review · Reviewer_gtAW · 2024-11-04

**Soundness:** 2
**Presentation:** 3
**Contribution:** 3
**Rating:** 6
**Confidence:** 2

**Summary:**

This paper presents a new method to model individual ways of understanding and interpreting medical images. In the filed of medicine, experts often see images differently based on their personal experiences and knowledge. Therefore, this paper aims to capture these unique perspectives by creating custom models for each doctor that reflect how they personally perceive similarities in medical images. To achieve this, the authors combine supervised learning for views image similarity, and unsupervised learning with an autoencoder to build a broader model of each doctor's cognitive pattern without requiring labels. In addition, the model uses triplet loss to help the system understand which images a doctor sees as more similar or different from each other. The authors conducted an extensive experiment with 121 doctors, asking them to judge the similarity between chest X-rays.

**Strengths:**

1.	The paper is well-structured, with clear explanations of the proposed framework and its components. The authors provide a thorough background, outlining the challenges in modelling person-specific cognitive similarities and the limitations of existing methods.
2.	The research is underpinned by a robust experimental design involving 121 clinical physicians, providing a substantial dataset for analysis. The methodology is meticulously detailed, ensuring reproducibility and transparency.  The integration of behavioural sampling to capture each participant's perception of similarity among chest X-ray images adds depth to the study, reinforcing the reliability of the findings.

**Weaknesses:**

1.	In Section 3.1, the authors implemented the measurement of cognitive similarity through a triangular arrangement of images, where physicians arrange images based on perceived closeness. However, this may lack depth in capturing nuanced interpretative differences. This approach does not account for context-dependent interpretation, such as how a physician might consider patient demographics or clinical history when assessing similarity. Therefore, the authors can benefit from incorporating more sophisticated cognitive tests or context-dependent tasks that could improve the understanding of the factors that influence these cognitive patterns. This added information could be used to fine-tune the embedding model.
2.	The authors use a convolutional autoencoder for CXR images and prove its efficiency. However, testing alternative architectures — like ViT (Dosovitskiy et al., 2020) — could provide insights into which architectures best capture complex cognitive similarities. Moreover, adding architectural flexibility or adaptivity within the model, perhaps by using modular components that can adjust based on data type, would make the framework more broadly applicable.
3.	The paper lacks comparisons with alternative embedding models that could serve as baselines. Without baselines, it is difficult to understand whether the proposed autoencoder with variable triplet loss truly excels over other methods.

Dosovitskiy, Alexey et al. “An Image is Worth 16x16 Words: Transformers for Image Recognition at Scale.” ArXiv abs/2010.11929 (2020): n. pag.

**Questions:**

1. Could you give a more detailed explanation of the variable triplet loss function? And why do you define such variable triplet loss and what mechanisms allow it to adapt to individual cognitive patterns?
2. While the study focuses on chest X-ray images, have you considered applying this approach to other medical imaging modalities, such as MRI or CT scans? If so, what adaptations would be necessary to accommodate the distinct characteristics of these modalities?
3. Could you elaborate on the decision to utilize CNNS for the autoencoder component instead of Transformer-based architectures? Given that Transformers have demonstrated effectiveness in capturing long-range dependencies and global context in image processing tasks, what were the considerations that led to favouring CNNs in this context? Additionally, how was the network architecture determined to ensure optimal performance in fine-grained image processing tasks?

---

> ### Author Response · Authors · 2024-11-18
> **We deeply appreciate your constructive review comments (Response 1)**
>
> We deeply appreciate your constructive review comments. We hope the responses below address your concerns effectively.
>
>
> 1. Concerns regarding sampling information (Weakness 1)
>
> We agree with the reviewer’s opinion. If our work were considered a practical medical application, our sampling methodology might not fully capture subtle interpretative differences. However, our primary goal is to bridge the gap between metric learning and modeling physicians’ similarity judgments. Our objective is not direct medical application, but rather using real-world data to test hypotheses in the context of cognitive science applications of metric learning.
>
> Specifically, we aim to demonstrate the feasibility of personalized embedding models using real-world data, not benchmark datasets, in realistic scenarios. To achieve this, we first validated our hypothesis in the simplest possible setting, as there is no evidence to date that personalized embedding modeling is feasible even in single-modality expert data.
>
> In our study, physician participants judged similarities based solely on CXR images, without additional clinical information. If personalized embedding can be achieved under these conditions, it could lead to future research incorporating contextual information or metadata into embedding models, as the reviewer suggested.
>
> This simple setup is key to demonstrating the generalizability of our approach. If we can model person-specific similarity using just images, without relying on domain-specific assumptions (e.g., medical histories), it could show that this framework applies to other expert domains, even in single-modality scenarios.
>
> We appreciate the reviewer’s insights and will consider them as we expand this research for medical domain applications.
>
>
> 2. Variable Triplet Loss
>
> Thank you for raising such an important question. Our Variable Triplet Loss is not an innovative development but rather a practical modification of the traditional triplet loss proposed in previous studies. In conventional machine learning, metric learning typically learns a general metric that averages similarity judgments across many data points. In contrast, for high-complexity data (e.g., medical images), individual similarity patterns can vary significantly, meaning that metrics should be sampled on an individual basis. However, it is difficult to sample many data points from a single individual, and there is high uncertainty in the information obtained from these samples. To address this issue of individual similarity modeling, we modified the traditional loss function to enhance its practical utility. Below, we compare the traditional triplet loss with our Variable Triplet Loss:
>
>
>
> Traditional Triplet Loss:  $L(A, P, N) = \max \left(  \left| f(A) - \hat{f(P)} \right|^2 -  \left| f(A) - \hat{f(N)} \right|^2 +margin, 0 \right)$
>
> Our Triplet Loss: $L(A, P, N) = \max \left( \alpha \left| f(A) - \hat{f(P)} \right|^2 - \beta \left| f(A) - \hat{f(N)} \right|^2, 0 \right)$
>
> For comparison, please note that the terms in our Equation (1) in the paper correspond to the traditional loss function as follows:
>
> C (closed) → P (positive), D (distant) → N (negative), A (anchor) →same, E() → f().
>
>
> Our loss function has two main advantages for learning individual embeddings:
>
>
> (1) Absence of margin term:
>
> In traditional triplet loss, there is an explicit margin term that forces the representation distance between positive samples to be closer than the distance between negative samples. However, our loss function does not use a margin but instead uses weighting terms 𝛼 and 𝛽 to encourage the representations of positive samples to be closer to the anchor than those of negative samples. However, unlike traditional loss, we do not enforce the representation distance between positive samples to be strictly closer than that of negative samples. This is significant because, unlike typical metric learning, where average trends are learned, individual similarity data for expert datasets is sparse and could reflect errors or outliers. For instance, when sampling similarity from 100 sets, 5-10 of these samples may not align with the individual's general tendency due to human error. In such cases, it is important not to force the model to learn outliers as part of the general tendency through a margin, but instead to ensure that data points that deviate from the overall trend are not included in the embedding model.
>
> (2) Constant embedding for Positive and Negative samples during training:
>
> In our triplet loss, the values of $\hat{f(P)}$ and $\hat{f(N)}$ are treated as constants calculated from the model of the previous epoch. In contrast, traditional triplet loss treats ${f(P)}$ and ${f(N)}$ as variables to be learned. Our setup is based on the observation that when triplet loss is combined with the reconstruction loss of an autoencoder, it is empirically more stable to treat $\hat{f(P)}$ and $\hat{f(N)}$ as constrants and only learn $f(A)$.

---

> ### Author Response · Authors · 2024-11-18
> **We deeply appreciate your constructive review comments (Response 2)**
>
> 3. Applicability to Other Modalities such as CT and MRI (Question 2)
>
> In our experiment, we did not introduce any special settings that are unique to Chest X-ray (CXR). For example, as we mentioned earlier, clinical information that doctors typically consider when interpreting CXR was not provided. Therefore, we believe that this approach can be applied to other modalities, such as MRI or CT, and could also be generally applicable to other non-medical image datasets. However, CT and MRI differ from X-ray images in that they have many more dimensions and are in voxel-based 3D data form. When interpreting CT or MRI scans, doctors do not consider the entire 2D tensor as with CXR, but instead focus on specific 3D lesions. Therefore, applying our framework to CT or MRI would be more practical if we consider the similarity of lesion-centered partial images rather than the entire image. Additionally, the quality and emphasis of the image can vary significantly depending on the image synthesis parameters, which can introduce strong biases in determining similarity. Therefore, controlling these parameters would need to be considered. Furthermore, unlike Chest X-rays, CT and MRI scans are often handled by physicians with different specialties, and some may not handle them at all. Thus, it would be important to control for subjects' specific specialties in experimental validation.
>
>
> 4. Decision to use CNN instead of Transformer and the design of our network architecture (Weakness 2 & Question 3).
>
> Thank you for the important question. We would like to emphasize two aspects regarding our decision to use CNN instead of Transformer.
>
> (1) First, unlike traditional metric learning, the amount of similarity information available for individual metric learning is very limited. In our experiment, each participant was sampled 500 times, and even with this small amount of data, it took more than 5 hours to process. To fit such small-scale sampling data, a relatively lightweight model architecture would be more suitable. Our goal was not to find the optimal architecture but to demonstrate the effect of combining unsupervised and supervised learning for individual metric learning. Therefore, we performed the analysis using the simplest architecture suitable for small datasets. In our future work, we plan to conduct performance comparison studies using various foundational models, such as transformers, to identify the optimal architecture.
>
> (2) Second, there is significant evidence that CNN representations overlap highly with human visual perceptual characteristics. (Jha, A., Peterson, J. C., & Griffiths, T. L. (2023). Extracting low‐dimensional psychological representations from convolutional neural networks. Cognitive science, 47(1), e13226; Lindsay, Grace W. "Convolutional neural networks as a model of the visual system: Past, present, and future." Journal of cognitive neuroscience 33.10 (2021): 2017-2031.) There is also evidence that CNNs can model human visual information processing at an abstract level (Kubilius, Jonas, et al. "Brain-like object recognition with high-performing shallow recurrent ANNs." Advances in neural information processing systems 32 (2019)). While transformers are known to be effective in capturing global context in image processing tasks, there was insufficient evidence to suggest that they could capture human like cognitive features effectively.
>
>
> After deciding to use CNN as our base model, we gave considerable thought to the detailed architecture. We were inspired by the Cornet model (Kubilius, Jonas, et al. "Brain-like object recognition with high-performing shallow recurrent ANNs." Advances in neural information processing systems 32 (2019)), which mimics the human visual information processing system. So we designed an encoder-decoder structure consisting of four layers. The specific parameters for each layer were determined empirically through several preliminary experiments to find near-optimal values.
>
>
>
> 5. Lack of comparison with alternative embedding models as baselines (Weakness 3)
>
> In the ablation study presented in Table 1 of the paper, we conducted a comparative experiment by removing the triplet loss and reconstruction loss individually. It is noteworthy that even without additional external information (such as human behavior data or labels), the combination of triplet loss and reconstruction loss resulted in synergistic performance improvements. The unsupervised learning introduced by the reconstruction loss typically involves the model finding features on its own; however, when combined with the triplet loss, it optimizes towards discovering cognitive features that differ from person to person.
>
> That said, we acknowledge the comment regarding the lack of baseline comparison models and will perform an analysis on a model that combines only the encoder with triplet loss. The results will be included in the upcoming revised version.

---

> > ### Comment · Reviewer_gtAW · 2024-11-29
> >
> > The authors' responses are appreciated, which has solved part of my concerns. However, after reading the other reviewers' comments, I decided to keep my score for now.

---

### Official Review · Reviewer_k73t · 2024-11-04

**Soundness:** 2
**Presentation:** 3
**Contribution:** 2
**Rating:** 5
**Confidence:** 4

**Summary:**

This paper addresses the challenge of modeling individual cognitive embeddings in expert domains, like medical data, where perceptions of features and similarities vary significantly among individuals. It proposes a novel approach that combines supervised learning on limited similarity sampling data with unsupervised autoencoder-based manifold learning to enhance the accuracy of person-specific psychological embeddings. The results from a large-scale study involving clinical physicians show that even with limited behavioral data, the proposed method effectively approximates these embeddings and improves similarity inference performance in complex domains.

**Strengths:**

1. The proposed method intriguingly combines supervised learning on small-scale similarity sampling data with unsupervised autoencoder-based manifold learning to approximate person-specific psychological embeddings.

2. The authors conducted a comprehensive experiment involving 121 clinical physicians, measuring their cognitive similarities using medical image data, which lends credibility to the results.

**Weaknesses:**

1. The paper's organization is unclear, making it difficult to follow. For instance, Figure 2 lacks sufficient detail and explanation regarding how the various modules interact with one another.
 2. Figure 1 and Equation 1, which illustrate the person-specific cognitive embedding modeling framework and the variable triplet loss function, lack detailed explanation. The authors should clarify how the proposed variable triplet loss function (Equation 1) innovatively captures individual cognitive similarity compared to standard triplet loss functions. A comparative analysis with traditional triplet loss in terms of mathematical formulation and expected outcomes would be beneficial.
3. The authors should conduct a sensitivity analysis on $\alpha$ and $\beta$ to demonstrate the robustness of the model concerning these critical parameters.
4. Figure 3 and Section 4.3 present the group-based similarity pattern analysis results. While the t-SNE visualizations illustrate the variability in similarity patterns among subjects, the paper lacks a statistical test to quantify the significance of these differences. The authors should include statistical validation, such as ANOVA or post-hoc tests, to confirm the variability of cognitive similarity patterns across different subjects. Additionally, the paper should discuss how these findings may generalize beyond the specific group of clinical physicians studied.

**Questions:**

The authors are required to address all my concerns carefully listed in the Weaknesses part.

---

> ### Author Response · Authors · 2024-11-18
> **Thank you for your valuable comments**
>
> Thank you for your valuable comments. Your review comments have greatly contributed to the improvement of the paper, and we will incorporate the feedback into the revised version, which will be uploaded in the next few days.
>
>
> 1. Organization of the Paper and Figure 2
>
> We apologize for the shortcomings in the organization of the main text and the presentation of Figure 2. Regarding Figure 2, the intention was not to depict different modules but rather to illustrate information received at different time points (previous epoch) from the same module. We will make this clarification in the revised version.
>
> Additionally, we recognize that certain parts of the paper’s organization are awkward, such as content that should belong in the Methods section being included in the Experiments section. These issues will be addressed in the revised version, and we kindly ask you to refer to the updated manuscript for these improvements.
>
>
> 2. Variable triplet loss
>
> Thank you for your comments. Please refer to the explanation we have provided below. We will incorporate the key points from the explanation into the main text of the paper and include the detailed aspects in the appendix.
> Our Variable Triplet Loss is not an innovative development but rather a practical modification of the traditional triplet loss proposed in previous studies. In conventional machine learning, metric learning typically learns a general metric that averages similarity judgments across many data points. In contrast, for high-complexity data (e.g., medical images), individual similarity patterns can vary significantly, meaning that metrics should be sampled on an individual basis. However, it is difficult to sample many data points from a single individual, and there is high uncertainty in the information obtained from these samples. To address this issue of individual similarity modeling, we modified the traditional loss function to enhance its practical utility. Below, we compare the traditional triplet loss with our variable Triplet Loss:
>
>
>
> Traditional Triplet Loss:  $L(A, P, N) = \max \left(  \left| f(A) - \hat{f(P)} \right|^2 -  \left| f(A) - \hat{f(N)} \right|^2 +margin, 0 \right)$
>
> Our Triplet Loss: $L(A, P, N) = \max \left( \alpha \left| f(A) - \hat{f(P)} \right|^2 - \beta \left| f(A) - \hat{f(N)} \right|^2, 0 \right)$
>
> For comparison, please note that the terms in our Equation (1) in the paper correspond to the traditional loss function as follows:
>
> C (closed) → P (positive), D (distant) → N (negative), A (anchor) →same, E() → f().
>
>
> Our loss function has two main advantages for learning individual embeddings:
>
>
> (1) Absence of margin term:
>
> In traditional triplet loss, there is an explicit margin term that forces the representation distance between positive samples to be closer than the distance between negative samples. However, our loss function does not use a margin but instead uses weighting terms 𝛼 and 𝛽 to encourage the representations of positive samples to be closer to the anchor than those of negative samples. However, unlike traditional loss, we do not enforce the representation distance between positive samples to be strictly closer than that of negative samples. This is significant because, unlike typical metric learning, where average trends are learned, individual similarity data for expert datasets is sparse and could reflect errors or outliers. For instance, when sampling similarity from 100 sets, 5-10 of these samples may not align with the individual's general tendency due to human error. In such cases, it is important not to force the model to learn outliers as part of the general tendency through a margin, but instead to ensure that data points that deviate from the overall trend are not included in the embedding model.
>
>
> (2) Constant embedding for Positive and Negative samples during training:
>
> In our triplet loss, the values of $\hat{f(P)}$ and $\hat{f(N)}$ are treated as constants calculated from the model of the previous epoch. In contrast, traditional triplet loss treats ${f(P)}$ and ${f(N)}$ as variables to be learned. Our setup is based on the observation that when triplet loss is combined with the reconstruction loss of an autoencoder, it is empirically more stable to treat $\hat{f(P)}$ and $\hat{f(N)}$ as constrants and only learn f(A).
>
>
> 3. Sensitivity analysis
>
> Thank you for your excellent comment. We are currently performing sensitivity analysis and will include the results in the appendix of the revised version.
>
>
> 4. Statistical validation of similarity pattern diversity
>
> We emphasize the diversity of similarity patterns among subjects and will conduct a statistical test to validate the randomness of this distribution. The results will be included in the revised version.
>
>
> We will inform you of the incorporated changes once the revised version is uploaded.

---

> > ### Comment · Reviewer_k73t · 2024-11-25
> >
> > I appreciate the authors' effort in rebuttal. Most of my concerns have been addressed.

---

> > > ### Author Response · Authors · 2024-11-25
> > >
> > > Dear Reviewer,
> > >
> > > I am pleased to inform you that the updated version of our manuscript has been uploaded. The new version incorporates your valuable feedback, and the following revisions have been made:
> > >
> > > 1. Unclear Organization:
> > > Section 2 on Related Work has been reorganized for better clarity, and Section 2.2 has been strengthened to provide a smoother transition into the discussion of personalized embedding modeling. In Section 4, the qualitative evaluation methods described have been moved to Section 3.5 to more clearly separate the methods and experimental results. Additionally, Figure 2 now clearly indicates that the differences represent time steps of the same module, rather than different modules, and unnecessary details that could have caused confusion have been removed.
> > >
> > > 2. Hyperparameter Sensitivity Analysis:
> > > The results of the sensitivity analysis have been updated in Appendix (A.7).
> > >
> > > 3. Statistical Analysis of Similarity Pattern Diversity:
> > > In Line 376, we have included the statistical analysis of the diversity of similarity patterns (multivariate runs test) along with the corresponding p-value.
> > >
> > > Please note that further updates to the manuscript may be made before the final deadline.
> > >
> > > Kind regards,

---

### Official Review · Reviewer_xoWe · 2024-11-06

**Soundness:** 2
**Presentation:** 3
**Contribution:** 2
**Rating:** 3
**Confidence:** 3

**Summary:**

This paper explores approximating person-specific cognitive embeddings in expert domains, where similarity perceptions vary between individuals.  The authors combine supervised learning on limited similarity data with unsupervised autoencoder-based manifold learning.  An experiment with clinical physicians and medical images demonstrates the feasibility of this approach.  The paper contributes a new method for modeling individual-level psychological embeddings, showing the potential of autoencoders in this context, and validating the use of variable triplet loss.

**Strengths:**

- The paper tackles a unique problem: approximating person-specific cognitive embeddings, particularly in domains with high inter-observer variability like medical image interpretation. This approach is a novel application of metric learning.

- The study includes an experiment with 121 clinical physicians, using real medical image data. This provides empirical support to their claims.

- The paper clearly outlines the methodology, including the triangular measurement framework for collecting behavioral data and the integration of supervised and unsupervised learning for embedding modeling.

**Weaknesses:**

- The core focus of the study leans more towards cognitive science and human-computer interaction, with limited novelty in terms of machine learning techniques. The use of standard autoencoder architectures and the absence of new metric learning algorithms may lessen its impact on the machine learning community.

- The study primarily focuses on medical image interpretation with limited exploration of the generalizability of the proposed approach to other domains. Further experiments on diverse datasets and tasks would strengthen the paper's contribution.

- The paper lacks a thorough analysis of the proposed method, especially concerning the convergence properties of the loss function and the interaction between triplet loss and manifold learning in autoencoders.

**Questions:**

- Could the problem of approximating person-specific embeddings be framed in the context of existing machine learning challenges like label noise or annotator disagreement? This could help position the work within a more familiar framework for the ML audience.

- Have you considered evaluating the approach on publicly available benchmark datasets for label noise or annotator disagreement? This would provide a point of comparison with existing methods and offer insights into the generalizability of your findings.

- How does the proposed approach scale with an increasing number of individuals and data points? Are there any considerations for improving the computational efficiency of the method?

- How can the insights from this study be used to develop personalized learning strategies for experts or improve human-AI collaboration in domains with high inter-observer variability?

---

> ### Author Response · Authors · 2024-11-18
> **We deeply appreciate the constructive questions and comments (Response 1)**
>
> We deeply appreciate the constructive questions and comments from the reviewer. Your feedback has been invaluable in helping us set a clear direction for improving our work. We hope the following responses effectively address your concerns.
>
>
> 1. Limited novelty in terms of machine learning  (Weakness 1 & 3)
>
> We agree with the reviewer’s comment regarding the perceived lack of novelty in our machine learning methodology. However, when submitting this paper, we clearly selected the "applications to neuroscience & cognitive science" field from the official topics for ICLR 2025 (Reference: https://iclr.cc/Conferences/2025/CallForPapers). Since the application of machine learning to cognitive science is an official topic for ICLR 2025, we believe that the lack of novel machine learning methodologies in this paper should not be considered a weakness. We wish to emphasize that our work is a novel application rather than a theoretical contribution to machine learning methods. There is a significant gap in the research on applying metric learning methods to individual metrics, and we are presenting a pioneering case for modeling individual metrics on expert data. This provides strong evidence for the scalability of metric learning methods in real-world data.
>
>
> Furthermore, regarding the reviewer's comment that our paper may have limited impact on the machine learning community, we believe that our work can still positively influence the field for the following reasons:
>
>
> (1) We have gathered similarity sampling data from 121 experts (clinical doctors), which required considerable time and effort, unlike experiments conducted on the general population. Expert-level behavioral sampling data in the human-in-the-loop machine learning field is rare, and we believe this data can be widely used for validating machine learning methodologies. We plan to make all the data publicly available through ICLR 2025. If the high-cost data collection is recognized as an independent contribution, it could motivate related research institutions to actively participate in data collection and sharing. We believe this will encourage more research utilizing high-cost data in the machine learning community.
>
>
> (2) One of our contributions is presenting a new application of autoencoders. We offer experimental evidence that the flexibility of unsupervised learning can be applied to cognitive science modeling. This may stimulate further research on autoencoders that control manifold learning, potentially advancing this line of study.
>
>
> 2. Exploration of Benchmark Dataset Application and Generalizability (Weakness 2 and Question 2)
>
> We greatly appreciate the important comment. We agree with the point that experimental validation using other datasets is necessary to rigorously demonstrate generalizability. Unfortunately, during the review period of this paper, it is physically difficult to conduct additional experiments with other datasets. However, given the purpose of our work, we approached the use of benchmark datasets with some caution. The reason is that many benchmark datasets tend to show little variation in the similarity patterns between individuals. In the case of benchmark datasets, there are conventional metrics for determining what data is perceived as similar or not. Even if individuals make judgments that deviate from these criteria for certain features, the overall trend remains consistent. This is because benchmark datasets typically have clear labels, and these labels serve as the key criteria for determining similarity metrics. While such benchmark datasets are reasonable for conventional metric learning (which aims to learn the average similarity metric across a population), it is uncertain whether they are suitable for person-specific metric learning. In contrast, expert data is more complex and uncertain, with highly varied individual similarity recognition patterns. Therefore, to explore the generalizability of our framework, experiments based on more complex data or expert-driven datasets might be necessary. We would like to respectfully note that conducting experiments with expert subjects and expert data requires significant costs, making it quite challenging to perform experiments across multiple expert domains simultaneously. We believe that if contributions from single expert experiments are acknowledged, it will encourage future researchers to carry out various behavior experiments targeting expert datasets.
>
> Additionally, while limited, we would like to emphasize that, in exploring generalizability, we independently used the CXR-A and CXR-B datasets, which have substantially different characteristics (i.e., different distributions) in terms of image properties. Furthermore, to carefully address the possibility of overclaiming our argument regarding generalizability, we are considering adding the qualifier ": focusing on medical images" to the title of our paper.

---

> ### Author Response · Authors · 2024-11-18
> **We deeply appreciate the constructive questions and comments (Response 2)**
>
> 3. The utility of person-specific embeddings in the context of machine learning frameworks (Question 1)
>
> First, one area where our framework can be applied in the context of existing machine learning is in active learning scenarios. In active learning, the problem of selecting unlabeled data for querying an oracle is cost-dependent. Therefore, if possible, querying the oracle about a dataset it deems dissimilar could be a strategy to extract diverse information within the cost constraints. By applying our idea, we can identify data distributions that the oracle deems similar, thus improving the efficiency of active learning.
>
> Second, in practical applications like Reinforcement Learning with Human Feedback (RLHF) for large language model (LLM) development, challenges arise when it is difficult to standardize or normalize human feedback. Our framework could be used to categorize and normalize feedback based on individual similarity patterns, providing a strategy to address this issue.
>
>
> 4. Scale with an increasing number of individuals and data points (Question 3)
>
> Modeling is performed independently for each individual, so as the number of individuals increases, the computational load increases proportionally. Each model is independent, and the performance of the model is defined at the individual model level, so there is no direct correlation between the number of individuals and performance. (We implemented 121 individual models for 121 subjects, and each model showed an average prediction accuracy of 68%.)
>
> An increase in the number of sampling data is expected to significantly contribute to overall performance improvement. The evidence for this is presented in Fig. 6(b). In simulations conducted with a number of data equivalent to human behavior experiments, we achieved prediction performance comparable to or slightly better than that of the human behavior experiments. However, as the number of sampling data increases, the simulation performance also increases proportionally. Despite this, increasing the number of samples to model individual similarity is practically difficult, so it will be more important to improve sampling efficiency or adopt approaches to enhance model performance (such as methods to reduce uncertainty).
>
>
> 5.	How can the insights from this study be used to develop personalized learning strategies for experts or improve human-AI collaboration? (Question 4)
>
> In the field of human-AI collaboration, the most challenging aspect is handling human uncertainty. Uncertainty is subjective but can be considered a type of function, and there is considerable evidence that human uncertainty can be modeled using machine learning (Yujin Cha and Sang Wan Lee. Human uncertainty inference via deterministic ensemble neural networks. In 35th AAAI Conference on Artificial Intelligence/33rd Conference on Innovative Applications of Artificial Intelligence/11th Symposium on Educational Advances in Artificial Intelligence, pp. 5877–5886. ASSOC ADVANCEMENT ARTIFICIAL INTELLIGENCE, 2021.)
>
> In other words, it is theoretically possible to model uncertainty by sampling it from individuals for specific data. However, the practical challenge in uncertainty modeling arises from the difficulty in approximating an individual’s abstracted representation space. For example, if a specific individual perceives certain data as similar, it is likely that their uncertainty about those data points will also be similar. This allows us to approximate uncertainty about a wide range of abstract data representation defined in the individual’s cognitive space using limited uncertainty sampling information. In doing so, we could assist in precision learning for humans using AI and apply it to classify cases where AI should query the human or delegate judgment.
>
> Moreover, if we could approximate an expert's cognitive embedding, we could use the inferred pattern to recommend specialists who can handle specific situations well. By quantifying the degree of similarity in perceptions of specific diseases, we could identify a highly specialized physician who knows a particular condition thoroughly.
>
> From a machine learning perspective, if there is a highly capable human expert, we might gain inspiration for implementing high-performance machine learning models by reconstructing the expert's embedding.
>
> If you have any additional questions or unresolved concerns, please let us know, and we are ready to provide detailed answers!

---

> > ### Comment · Reviewer_xoWe · 2024-11-26
> > **Response to authors**
> >
> > I appreciate the authors' detailed responses to my previous review. They've clarified the motivation behind their work and highlighted the potential contributions, particularly concerning the expert-level dataset and the novel application of autoencoders.
> >
> > However, I still believe that demonstrating the generalizability of the proposed method is crucial for its acceptance. External validation on benchmark datasets, even if those datasets aren't perfectly suited for person-specific learning, would significantly strengthen the paper. Additionally, a more in-depth analysis of how the method connects to and potentially improves existing active learning techniques would solidify its position within the machine learning literature. While I maintain my current score, I am open to reconsidering it if the authors can convincingly address these remaining concerns.

---

> > > ### Author Response · Authors · 2024-11-26
> > >
> > > Thank you for your comments.
> > > We hope the following response addresses your concerns.
> > >
> > > First, we also recognize the importance of exploring the potential generalization of our framework across diverse domains. As this study is grounded in human behavior experiments, external validation using benchmark datasets requires additional time and resources. Given the challenges of promptly recruiting participants and conducting experiments, we plan to address this aspect in future research.
> > > While we fully appreciate the reviewer’s perspective, we respectfully request your understanding that, in machine learning research involving human behavior experiments, studies utilizing general benchmark datasets and those leveraging specialized datasets each hold unique significance. In this context, we believe that studies like ours, which employ only specialized data, can also contribute meaningfully to the fields of machine learning and cognitive science.
> > > It is worth noting that human behavior experiment research involves substantial costs and time, beyond those required for the modeling process itself. Consequently, conducting experimental research that simultaneously uses both benchmark and specialized datasets is highly challenging and, to the best of our knowledge, unprecedented. We hope that recognizing the independent contributions of study conducted with either general or specialized datasets will support and encourage the development of high-risk (and resource-intensive) human behavior experiment-based study in the future.
> > >
> > >
> > > Second, in traditional machine learning frameworks, active learning assumes a perfect oracle, with oracle annotations considered error-free. However, in real-world active learning scenarios, the oracle is noisy; that is, human experts acting as oracles may have uncertainty, leading to potential annotation errors. Traditional active learning considers uncertainty solely from the model’s perspective, but to optimize model performance under limited budgets, it is advantageous to query data points with high uncertainty from the model’s perspective but low uncertainty from the oracle’s perspective. Unlike the model, it is difficult to measure or estimate the oracle’s uncertainty for all candidate data points. Therefore, a practical approach is to measure the oracle’s uncertainty for a subset of samples and then model this uncertainty. Just as the model’s uncertainty is a function in the representation space (embedding) rather than the original high-dimensional space, modeling human oracle uncertainty requires understanding the psychological embedding that represents the human oracle’s similarity. In this context, similar data points in the psychological embedding space share similar uncertainty. Since individuals exhibit different similarity patterns, their psychological representation spaces—and consequently their uncertainty models—also differ.
> > >
> > > In summary, in active learning, it is advantageous for the model to query data points with low oracle uncertainty. Since measuring oracle uncertainty for all data points is not feasible, modeling is necessary. This, in turn, requires oracle uncertainty measurements for sample data and a psychological embedding model. Therefore, reconstructing a psychological embedding space that reflects person-specific similarity patterns can enhance the accuracy and efficiency of human-in-the-loop processes such as active learning.
> > >
> > > Please let us know any additional comments or questions.
> > > Thank you.

---

### Meta-Review · Area_Chair_ESfG · 2024-12-18

**Metareview:**

This study explores individual psychological embeddings in expert domains, such as medicine, where cognitive similarities vary among professionals. This paper initially received mixed scores, and the authors did not fully address the reviewer's concern. Specifically, there are two critical issues that remain unresolved:

1. **Lack of External Validation:** The paper does not include external validation using benchmark datasets, which is essential to assess the generalizability and robustness of the proposed method. This absence raises questions about the applicability of the findings in broader contexts.

2. **Technical Issues and Required Revisions:** The reviewers identified several technical shortcomings that need to be corrected. Addressing these would necessitate a comprehensive rewrite of the paper, which currently renders it unsuitable for publication in its existing form.

In light of these concerns, I recommend rejecting this submission. I hope the reviewers' comments improve the quality of this paper. The authors are strongly encouraged to address these points thoroughly before resubmitting.

**Additional Comments On Reviewer Discussion:**

My decision is based on the following key issues:

- reviewer `xoWe ` mentioned that external validation is necessary on benchmark datasets, and how the proposed method improves Sota active learning techniques remains unclear.

- Reviewer` tueP` mentioned that a significant rewrite of the work should be required to evaluate it as a new manuscript to clarify the novelty of the proposed method, the experiment significance, the justification for modeling individual data using embedding, etc.

Although this paper presents an interesting idea, the current version is still too immature to be polished. I have to recommend rejecting this paper for now. However, the authors are strongly encouraged to include the response in the main paper, systematically reorganize it, and then submit it to a future conference or journal.

---

### Decision · Program_Chairs · 2025-01-22

Reject